# A framework for detecting and predicting highway traffic anomalies via multimodal fusion and heterogeneous graph neural networks

Shaowei Sun[1,2*], Mingzhou Liu[1]

**1** School of Mechanical Engineering, Hefei University of Technology, Hefei, Anhui,China, **2** Intelligent Transportation Engineering Institute, Anhui Transport Consulting & Design Institute Co.,Ltd, Hefei, Anhui, China

* 2020020003@mail.hfut.edu.cn

## Abstract

This paper presents a novel framework for detecting and predicting abnormal traffic events on highways. Current traffic monitoring systems often rely on single data sources, which limits their detection accuracy and robustness in complex environments. To address these limitations, we propose a framework based on multimodal deep fusion and heterogeneous graph neural networks (HGNNs), incorporating an Ensemble Contrastive Pessimistic Likelihood Estimation (CPLE) algorithm to optimize performance. The framework integrates static and dynamic traffic data, such as video images, traffic flow, vehicle speed, and tunnel weather conditions. Experimental results demonstrate that the model performs well in various scenarios, showing significant improvement in accuracy and stability over existing models like AGC-LSTM and AttentionDeepST. For instance, the proposed MHGNN-CPLE model achieves superior accuracy and F1 score in static detection tasks while maintaining high accuracy under different noise levels in dynamic detection scenarios. This study provides a significant advancement in traffic event analysis by effectively combining multimodal data and leveraging HGNNs to capture complex spatiotemporal dependencies.

## Introduction

With the acceleration of urbanization, the efficient management of highway traffic has become increasingly challenging [1]. The real – time detection and prediction of abnormal traffic events, such as traffic accidents, congestion, and road condition changes caused by adverse weather, are particularly critical [2]. These abnormal traffic events can significantly impact the efficiency and safety of road traffic. However, most current traffic monitoring systems rely on a single data source, such as camera surveillance, radar, or sensors [3]. This approach often neglects the multimodal characteristics of traffic events, resulting in insufficient detection accuracy and poor robustness in complex environments [4–5].

**Data availability statement:** All relevant data for this study are publicly available from the GitHub repository (https://github.com/ShaoweiSun/A-framework-for-detecting-and-predicting-highway-traffic-anomalies-).

**Funding:** This work was supported by the Science and Technology Project of Housing and Urban-Rural Construction of Anhui Province, Grant No. 2022-YF044. The funders had no role in study design, data collection and analysis, decision to publish, or preparation of the manuscript.

**Competing interests:** The authors have declared that no competing interests exist.

The detection of abnormal traffic events on highways is critical for ensuring traffic efficiency and safety. Traditional methods, which rely on single data sources such as traffic flow monitoring, speed detection, and road surveillance videos, have shown limitations in accuracy and robustness [6–9]. These methods, often using rule-based engines or statistical models, are significantly affected by external environmental changes and data noise. Rule-based detection systems, while simple to implement, fail to respond promptly and accurately to complex traffic events like sudden accidents and adverse weather. Machine learning techniques, though promising, still struggle to effectively integrate different types of data, leading to limited predictive performance in complex traffic scenarios. Previous studies have attempted to address some of these issues, but none have comprehensively integrated multimodal data with heterogeneous graph neural networks (HGNNs) to capture complex spatio-temporal dependencies. To fill this gap, we propose a novel framework that combines multimodal deep fusion and HGNNs, enhanced by the Ensemble Contrastive Pessimistic Likelihood Estimation (CPLE) algorithm, to improve the accuracy and reliability of detecting and predicting abnormal traffic events.

To address these limitations, an advanced and comprehensive approach that can effectively handle the complexity and multimodal nature of traffic data is required. The integration of multimodal data provides a more comprehensive view of traffic scenarios, capturing various aspects of traffic conditions that cannot be obtained from a single data source. By combining multimodal data such as video images, traffic flow, vehicle speed, and tunnel weather conditions, we can better understand the underlying patterns and dependencies in traffic events. Moreover, heterogeneous graph neural networks (HGNNs) have shown great potential in modeling complex spatio-temporal dependencies and handling diverse data types. The Ensemble Contrastive Pessimistic Likelihood Estimation (CPLE) algorithm further enhances the model's robustness and accuracy by optimizing the detection and prediction process based on sample differences. Therefore, the proposed framework, which integrates multimodal deep fusion and HGNNs with the CPLE algorithm, aims to provide a more accurate and reliable solution for detecting and predicting abnormal traffic events on highways.

The detection of traffic flow anomalies is complex and closely connected to numerous factors like weather, road conditions, and equipment performance [10]. For example, Khan and Ahmed presented a deep learning approach to predict the severity of highway accidents in adverse weather, while Wang et al. used road sensor data to predict highway icing occurrence times [11]. These studies show that combining multimodal data with deep learning methods is highly promising for detecting and predicting abnormal traffic events [12].

Our proposed framework aims to improve the accuracy and reliability of detecting and predicting abnormal traffic events. The framework effectively integrates multimodal data and leverages HGNNs to capture complex spatiotemporal dependencies. The Ensemble Contrastive Pessimistic Likelihood Estimation (CPLE) algorithm is integrated to address the common challenge of uncertainty and imbalance in anomaly prediction. Unlike standard loss functions that treat all samples uniformly,

CPLE dynamically adjusts sample weights based on their similarity and prediction errors. By emphasizing the differences between samples, CPLE enhances the model's ability to recognize anomaly events. Its objective function is designed to minimize the contrastive pessimistic likelihood, thereby improving the detection accuracy of abnormal traffic events even in imbalanced datasets.

The main contributions of this paper are as follows:

(1) We present a novel framework integrating multimodal deep fusion and HGNNs for highway traffic abnormal event detection and prediction. It overcomes the limitations of single-data-source traditional methods by combining diverse data like images, point clouds, sound, and temperature.

(2) We propose a new multimodal data fusion mechanism. Through data preprocessing and feature extraction algorithms, it effectively fuses heterogeneous data, ensuring seamless integration and collaboration among different modal information. This mechanism comprehensively captures traffic flow changes, offering strong support for abnormal event identification in complex environments.

(3) Experimental results validate the model's high accuracy and stability in detecting and predicting various traffic abnormal events. Compared with existing models like AGC-LSTM and AttentionDeepST, the proposed MHGNN-CPLE model shows superior performance in static detection tasks and good robustness in dynamic detection scenarios.

## Related work

### Traditional methods for detecting abnormal traffic events on highways

Traditional methods for detecting abnormal traffic events primarily rely on single data sources such as traffic flow monitoring, speed detection, and road surveillance videos [13]. These methods often employ rule-based engines or statistical models to assess traffic conditions. However, their precision and reliability are significantly affected by external environmental changes and data noise, leading to suboptimal detection accuracy and adaptability in practical applications [14]. For example, rule-based detection systems, despite being simple and easy to implement, often fail to respond promptly and accurately to complex traffic events like sudden accidents and adverse weather. Although machine learning techniques have shown some promise in recent years, most methods still struggle to effectively integrate different types of data, resulting in limited predictive performance when dealing with complex traffic events. The application of machine learning methods generally requires a large amount of training data and places high demands on data preprocessing and feature extraction. Han et al. (2022) proposed an artificial neural network-based data cleaning framework for processing highway asphalt pavement detection data, which can effectively eliminate noisy data and improve the accuracy of subsequent models [15]. However, it still faces challenges in addressing data heterogeneity and dynamism in traffic abnormal event detection.

### Application of graph neural networks in traffic prediction

Graph Neural Networks (GNNs) have emerged as a powerful tool for managing complex spatiotemporal data in traffic flow prediction and anomaly detection. They effectively capture spatial dependencies and their dynamic changes in traffic flow, especially in transportation networks, due to their superior information transmission capabilities. As research progresses, heterogeneous graph neural networks (HGNNs) have gained significant attention for their ability to handle diverse nodes and edges, making them highly effective for multimodal data integration and complex traffic event modeling. Existing applications of GNNs in traffic flow prediction have shown promising results. For instance, Niu et al. (2022) used a deep learning method based on gantry toll samples to analyze spatiotemporal traffic flow patterns [16]. Zhang et al. (2022) applied deep learning for information security event inference and tracking in highway networks [17]. HGNNs have also demonstrated unique strengths in traffic anomaly detection. Zhou et al. (2023)

introduced a visual-based anomaly trajectory detection framework that combines GNNs with computer vision for real-time traffic event detection and alerting [18]. Elghaish et al. (2022) proposed a CNN model for detecting and classifying highway cracks, leveraging the structural advantages of GNNs [19]. Additionally, progress has been made in traffic data stream anomaly detection using GNNs. Guo et al. (2023) developed an ETC data anomaly detection method based on an enhanced DTW algorithm [20], while Zhong et al. (2022) created a deep learning-based high-way data flow risk monitoring system [21]. However, these studies often focus on specific aspects or data types. This study introduces a comprehensive multimodal framework that integrates various data sources (images, point clouds, sound, and temperature) to improve the detection and prediction of abnormal traffic events. This approach enhances prediction robustness and accuracy and addresses previous methods' limitations. In summary, with the continuous development of GNNs and their variants, their application prospects in traffic prediction and abnormal event detection are broad. HGNNs, in particular, have shown remarkable proficiency in processing intricate multimodal data and traffic events, providing novel concepts and technological foundations for future traffic flow prediction and intelligent transportation systems development.

## Comparative analysis of existing models

Recent studies have significantly advanced the field of traffic prediction and abnormal event detection by integrating various methodologies and applying cutting-edge technologies. Zhang et al. (2024) proposed a physics-informed deep learning framework that combines traffic flow models with computational graph methods, achieving improved accuracy in traffic prediction tasks [22]. Zhang et al. (2024) further extended this approach by developing a physics-guided deep learning model for short-term origin-destination demand prediction in urban rail transit systems, demonstrating robust performance under pandemic conditions [23]. Zhang et al. (2025) introduced the EF-former model for short-term passenger flow prediction during large-scale events in urban rail transit systems, leveraging the power of transformers to capture complex spatiotemporal dependencies [24]. Additionally, Zhang et al. (2025) presented a multi-frequency spatial-temporal graph neural network for short-term metro origin-destination demand prediction during public health emergencies, further advancing the application of graph neural networks in traffic prediction [25]. Qiu et al. (2025) proposed a spatial–temporal multi-task learning method for predicting passenger inflow and outflow in urban rail transit systems, particularly during holidays [26]. Sharafian et al. (2025) explored fuzzy adaptive control strategies for consensus tracking in multi-agent systems with incommensurate fractional-order dynamics, offering innovative solutions for managing traffic in uncertain and dynamic environments [27]. Sharafian et al. (2025) also investigated the resilience of adaptive RBF neural networks against deception attacks in consensus tracking control of incommensurate fractional-order power systems, highlighting the importance of robustness in intelligent transportation systems [28]. Ali et al. (2025) proposed an energy-efficient resource allocation strategy for urban traffic flow prediction in edge-cloud computing environments, optimizing resource usage while maintaining prediction accuracy [29]. Ali et al. (2024) introduced a resource-aware multi-graph neural network for urban traffic flow prediction in multi-access edge computing systems, further advancing the application of graph neural networks in traffic prediction [30]. Ali et al. (2025) presented an attention-driven spatio-temporal deep hybrid neural network for traffic flow prediction in transportation systems, effectively capturing complex spatial and temporal dependencies in traffic data [31]. Ali et al. (2021) proposed a data aggregation-based approach to exploit dynamic spatiotemporal correlations for citywide crowd flows prediction in fog computing, providing valuable insights into understanding and predicting traffic patterns in complex urban environments [32]. To better illustrate the advantages of our proposed framework, Table 1 provides a comprehensive comparison of existing highway abnormal event detection models with our method. The comparison covers various aspects, including model type, data types used, spatial-temporal dependency modeling, and performance metrics. The results highlight the superiority of our multimodal deep fusion and heterogeneous graph neural network (HGNN) framework in terms of accuracy, robustness, and adaptability to complex traffic scenarios.

 

**Table 1. Comparison of highway abnormal event detection models.**

| Model | Model Type | Data Types | Spatial-Temporal Dependency Modeling | Accuracy | Robustness | Adaptability |
|---|---|---|---|---|---|---|
| AGC-LSTM | Graph Convolutional LSTM | Traffic flow, vehicle speed | Limited | 0.965 | Medium | Low |
| AttentionDeepST | Deep Aggregation with Attention | Traffic flow, vehicle speed | Limited | 0.960 | Medium | Low |
| Physics-Informed DL (Zhang et al., 2024) | Physics-Informed Deep Learning | Traffic flow, vehicle speed | Moderate | 0.920 | High | Medium |
| EF-former (Zhang et al., 2025) | Transformer-based | Traffic flow, passenger demand | Advanced | 0.940 | High | Medium |
| Proposed MHGNN-CPLE | Multimodal Deep Fusion with HGNN and CPLE | Video images, traffic flow, vehicle speed, tunnel weather | Advanced | 0.980 | High | High |

## System design and model construction

In this research, a framework for detecting and predicting abnormal traffic events on highways based on multimodal deep fusion and heterogeneous graph neural networks (HGNNs) is constructed. The aim is to overcome the limitations of traditional methods and enhance the accuracy and robustness of detection and prediction. This framework mainly encompasses three core components: data collection and fusion, model development and evaluation, and model optimization. The overall architecture is shown in Fig 1.

## Design of multimodal data acquisition and preprocessing methods

This study involves four key types of multimodal data: images, point clouds, sound, and temperature data. To integrate these diverse data types effectively, a hybrid fusion strategy is employed. This strategy combines early fusion, where raw data from different modalities are synchronized and preprocessed together, with late fusion, where features extracted from each modality are weighted and combined at a later stage. Synchronization of modalities is achieved through time-stamping and alignment algorithms to ensure data from different sources corresponds to the same time frame. Weighting of each modality is dynamically adjusted based on historical data analysis, which assesses the quality and relevance of each data type to traffic anomaly detection.

Image data is obtained in real-time through highway cameras. Advanced computer vision algorithms, particularly convolutional neural networks (CNN) and region-based convolutional neural networks (R – CNN), are utilized to identify crucial elements within traffic scenes. In the preprocessing stage of image data, steps such as denoising (using techniques like Gaussian filtering or median filtering), scale normalization, and standardization are carried out. Additionally, components like multi – head attention and weighted graph convolution networks are employed to enhance the model's performance in complex traffic scenarios.

Point cloud data, provided by LiDAR sensors, captures the three – dimensional spatial information of roads and vehicles. The key to data preprocessing is to align and fuse point cloud data from different perspectives into a consistent 3D model. This study adopts a point cloud registration method based on the ICP (Iterative Closest Point) algorithm. The matching and fusion process of point clouds can be represented by the following mathematical model:

$$P_{fused} = \sum_{i=1}^{n} P_i \cdot w_i \tag{1}$$

where $P_{fused}$ is the fused point cloud data, $P_i$ is the point cloud data collected by the $i$-th sensor, and $w_i$ is the weighting coefficient of each sensor. A fusion strategy based on distance and angle weighting is applied to enhance the spatial information expression ability of point cloud data.

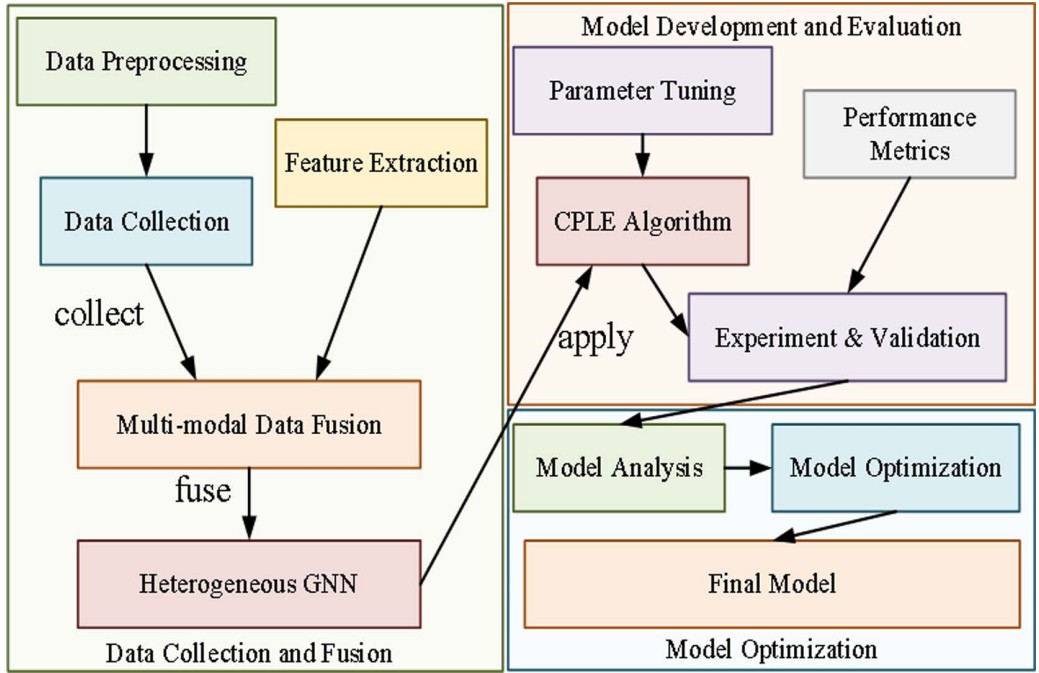

**Fig 1. Technical process of this article.**

Sound data, collected through sound sensors installed on the road, undergoes significant changes during traffic abnormal events. The preprocessing of sound data includes removing background noise, extracting key frequency features, and converting sound signals into frequency domain signals using Fourier transform. The preprocessing model for sound data can be represented as:

$$S_{cleaned} = F^{-1}(\mathcal{F}(S) \cdot H(f)) \tag{2}$$

In this equation, $S_{cleaned}$ refers to the denoised frequency domain signal obtained after processing. F symbolizes the Fourier transform operation that converts the signal from the time domain to the frequency domain, while S is the original sound signal captured by the sensors. H(f) denotes a frequency domain filter designed to eliminate noise components. Finally, $F^{-1}$ represents the inverse Fourier transform that converts the processed signal back to the time domain. Methods like frequency domain filtering and wavelet transform are used to eliminate irrelevant noise and improve the accuracy of sound signals.

Temperature data, collected through temperature sensors on the road, reflects temperature changes on the road surface. When processing temperature data, interpolation is first performed on the raw data to fill in missing values, and then linear interpolation is used to smooth the data and reduce its instability. Denoising is also carried out to remove interference caused by environmental factors. To ensure the consistency of multimodal data during fusion, all collected data undergoes unified standardization and normalization processing. The standardization method is implemented through Z-score normalization to unify feature scales and address domain discrepancies across heterogeneous data sources. For each modality, raw data X is transformed by subtracting the mean µ and dividing by the standard deviation σ, expressed as:

$$X_{norm} = \frac{X - \mu}{\sigma} \tag{3}$$

Where $X$ is the original data, $\mu$ is the mean, and $X_{norm}$ is the standard deviation.

A weighted fusion strategy is adopted, where the feature vectors of different modalities are weighted and merged to obtain the fused feature vectors of multiple modalities. The process can be expressed as:

$$x_{final} = \sum_{i=1}^{n} w_i \cdot x_i \tag{4}$$

Where $w_i$ is the weight of the $i$-th modality, reflecting its contribution to traffic anomaly event detection. Video data undergo a dual normalization process: pixel values are first scaled to [0,1] via min-max normalization to mitigate illumination variations, followed by Z-score normalization with $\mu = 0.5$ and $\sigma = 0.2$. For LiDAR point clouds, spatial coordinates are centered by subtracting the centroid mean ($\mu x$, $\mu y$, $\mu z$) and scaled by the standard deviation of their spatial distribution ($\sigma x$, $\sigma y$, $\sigma z$) to decouple positional biases. Sound signals in the frequency domain are normalized by the maximum amplitude observed in the training set ($\mu = 0$, $\sigma = \max(|S(f)|)$), ensuring consistent magnitude ranges. Temperature data are linearly scaled to [−1, 1] based on historical extreme values ($\mu = 0$, $\sigma = 15°C$) to encapsulate environmental variations.

Post-normalization, cross-modal calibration is performed: temporal alignment ensures all modalities share identical 5-minute time windows, while outliers (e.g., sensor malfunctions) are clipped to $\mu \pm 3\sigma$ to prevent skewed distributions. This rigorous standardization protocol preserves the statistical integrity of each modality while enabling seamless fusion for downstream tasks. To present the logic of multimodal data acquisition and preprocessing more clearly, we write the algorithm as pseudocode shown in Algorithm 1.

## Algorithm 1. Multimodal data acquisition and preprocessing

```
Input: Highway camera data, LiDAR point cloud data, road sound sensor data, road temperature sensor data
Output: Fused and preprocessed feature vectors
// Image data processing
1: image_data = GetRealTimeImageDataFromHighwayCameras()
2: denoised_image = DenoiseImage(image_data)
3: normalized_image = NormalizeScale(denoised_image)
4: enhanced_image_features = EnhanceFeatures(normalized_image)
// Point cloud data processing
5: point_cloud_data = GetPointCloudDataFromLiDAR()
6: aligned_point_clouds = AlignPointClouds(point_cloud_data)
7: fused_point_cloud = FusePointClouds(aligned_point_clouds)
// Sound data processing
8: sound_data = GetSoundDataFromRoadSensors()
9: cleaned_sound = RemoveBackgroundNoise(sound_data)
10: frequency_domain_signal = FourierTransform(cleaned_sound)
11: filtered_signal = FrequencyDomainFiltering(frequency_domain_signal)
// Temperature data processing
12: temperature_data = GetTemperatureDataFromRoadSensors()
13: interpolated_temperature = InterpolateMissingValues(temperature_data)
14: smoothed_temperature = SmoothData(interpolated_temperature)
15: denoised_temperature = Denoise(smoothed_temperature)
// Standardization and normalization of all data
16: standardized_image = Standardize(enhanced_image_features)
17: standardized_point_cloud = Standardize(fused_point_cloud)
18: standardized_sound = Standardize(filtered_signal)
19: standardized_temperature = Standardize(denoised_temperature)
// Weighted fusion of multimodal data
20: weights = CalculateWeightsBasedOnHistoricalData()
21: fused_feature_vectors = WeightedFusion([standardized_image, standardized_point_cloud, standard-
ized_sound, standardized_temperature], weights)
22: return fused_feature_vectors
```

## Heterogeneous graph neural network model design

Heterogeneous Graph Neural Networks (HGNNs) play a central role in the detection and prediction framework of abnormal traffic events on highways, as shown in Fig 2. This model design fully exploits the heterogeneous characteristics of different types of data in traffic flow and effectively fuses and learns information through multi – level graph convolutional networks (GCNs). Traffic data has strong spatiotemporal dependencies, and the relationships between different modal data are complex and diverse. HGNNs can efficiently transmit and learn information based on the characteristics of different types of nodes and edges.

In this study, the nodes of the graph represent multimodal data sources such as traffic flow, vehicle speed, accident information, and environmental temperature, while the edges represent the interrelationships between different data sources. A multi – level graph convolutional network architecture is designed to adaptively update nodes in each layer and effectively fuse feature information from different modal data sources. The update of each node is weighted and averaged based on its neighboring node information and edge weights. The specific process is described by the following formula:

$$h_i^{(l+1)} = \sigma\left(\sum_{j \in N(i)} \frac{1}{c_{ij}} W^{(l)} h_j^{(l)} + b^{(l)}\right)$$

(5)

where $h_i^{(l+1)}$ represents the hidden state of the $i$-th node in the $(l+1)$-th layer, $N(i)$ is the set of neighboring nodes of the $i$-th node, $h_j^{(l)}$ is the hidden state of the $j$-th neighboring node in the $l$-th layer, $W^{(l)}$ is the weight matrix of the $l$-th layer, $b^{(l)}$ is the bias term, $\sigma$ is the activation function, and $c_{ij}$ is the normalization coefficient of the edge.

Moreover, to further enhance the model's perception ability towards heterogeneous data sources, Weighted Graph Convolutional Networks (Weighted GCNs) are adopted. By introducing a weighting mechanism between different modal

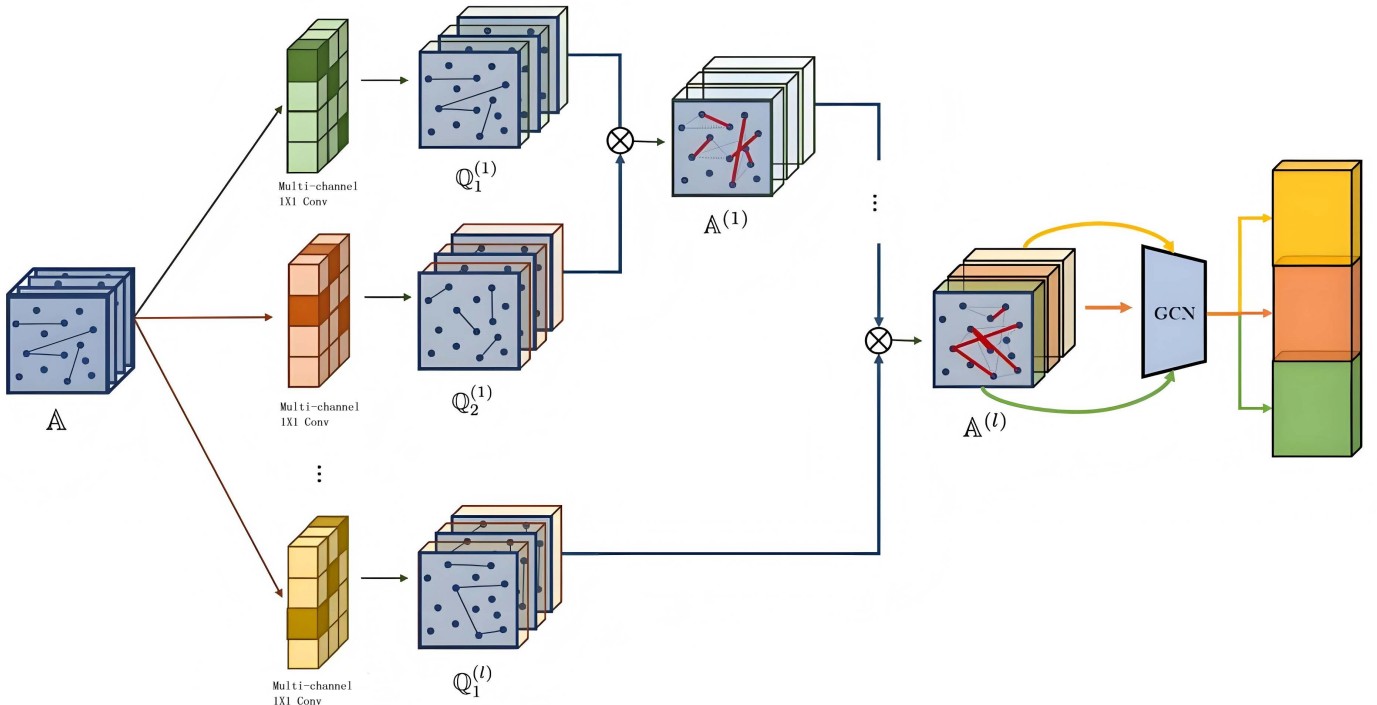

**Fig 2. Heterogeneous graph neural network structure.**

data, certain data sources can have a greater impact on the model's prediction results during the training process. The weighted strategy is modeled using the following formula:

$$x_i^{(l)} = \sum_{j \in N(i)} \alpha_{ij} W^{(l)} x_j^{(l)} + b^{(l)}$$

(6)

where $\alpha_{ij}$ is the weighting coefficient between $i$ and $j$ nodes, which can be dynamically adjusted according to the strength of the relationship between nodes.

For the fusion of multimodal data, a weighted fusion strategy based on the attention mechanism is employed, which can adaptively adjust the influence of different modal data in the process of information transmission. The weight of each modality is calculated using the following formula:

$$\omega_i = \frac{exp(a_i)}{\sum_{i=1}^{n} exp(a_i)}$$

(7)

where $a_i$ is the attention score representing the th modality, and is the weight of that modality. To improve the prediction accuracy of the model, the Contrastive Pessimistic Likelihood Estimation (CLL) algorithm is combined to optimize the model parameters during the training process. This algorithm enhances the recognition ability of abnormal events based on the relative differences between samples. The objective function of CLL can be expressed as:

$$\mathcal{L}_{CLL} = -\sum_{i=1}^{n} log \left( \frac{exp(z_i)}{\sum_{j=1}^{n} exp(z_j)} \right)$$

(8)

where $z_i$ is the prediction result of the input data in the current model, and $\mathcal{L}_{CLL}$ optimizes the accuracy of the model in predicting abnormal events by minimizing the objective function. Through the design of this heterogeneous graph neural network, it can comprehensively capture the complex relationships between different modal data and adaptively adjust the contributions of each modality in event detection through weighting and attention mechanisms, thereby improving the model's performance in predicting abnormal traffic events on highways. Combined with the pessimistic likelihood estimation algorithm, the entire framework can accurately and quickly identify potential traffic anomalies in dynamic traffic environments, providing timely warning and decision support for traffic management.To clearly illustrate the design of the heterogeneous graph neural network model, we express the algorithm in pseudocode as presented in Algorithm 2.

## Algorithm 2. Heterogeneous graph neural network model

```
Input: Multimodal data features, number of layers L
Output: Model predictions
// Initialize graph nodes and edges
1: nodes = InitializeNodes(multimodal_data_features)
2: edges = InitializeEdges(nodes)
// Initialize model parameters
3: for l=1 to L do
4:     W⁽ˡ⁾ = InitializeWeightMatrix()
5:     b⁽ˡ⁾ = InitializeBiasTerm()
6: end for
// Forward propagation
7: for l=1 to L do
8:     for each node i in nodes do
9:     hᵢ⁽ˡ⁺¹⁾ = ActivationFunction(
10:    Sum over j in N(i) of (1/ cᵢⱼ) * W⁽ˡ⁾ * hᵢ⁽ˡ⁺¹⁾ + b⁽ˡ⁾
```

```
11:    )
12: end for
13: end for
// Weighted graph convolution
14: for each node i in nodes do
15: x_i^(1) = Sum over j in N(i) of (α_ij * W^(1) * x_j^(1) + b^(1))
16: end for
// Multimodal data fusion with attention mechanism
17: attention_scores=CalculateAttentionScores(multimodal_data_features)
18: weights = CalculateWeightsBasedOnAttention(attention_scores)
19: fused_features = WeightedFusion(multimodal_data_features, weights)
// Combine with Contrastive Pessimistic Likelihood Estimation (CLL)
20: loss = CalculateCLLLoss(fused_features, true_labels)
21: UpdateModelParameters(loss)
22: predictions = MakePredictions(fused_features)
23: return predictions
```

## Integrated comparison pessimistic likelihood estimation (CPLE) algorithm design

To further optimize the inference process of the CPLE algorithm, multiple mathematical models are introduced to comprehensively adjust the sample weighting mechanism, loss function, and optimization strategy, aiming to improve the accuracy and robustness of anomaly event detection. The weighting coefficients of the samples are dynamically adjusted based on sample similarity. The $i$-th sample's weighting coefficient is updated based on the difference between the predicted results $z_i$ and those of neighboring samples $z_j$. The specific update formula is:

$$\omega_i^{(norm)} = \frac{\omega_i}{\sum_{i=1}^{n} \omega_i}$$

(9)

where $n$ is the total number of samples, $\omega_i$ is the original weighting coefficient of each sample, and $\omega_i^{(norm)}$ is the normalized weighting coefficient, ensuring that the total weight of all samples is 1. On the basis of updating the weighting coefficients, the loss function of the CPLE algorithm is adjusted according to the weight of each sample. Assuming the loss function is $\mathcal{L}$, the weighted loss function is:

$$\mathcal{L}_{weight} = \sum_{i=1}^{n} \omega_i^{(norm)} \cdot \mathcal{L}(y_i, \hat{y}_i)$$

(10)

where $\mathcal{L}(y_i, \hat{y}_i)$ is the standard loss function (such as cross – entropy loss), $y_i$ is the true label of the $i$-th sample, $\hat{y}_i$ is the predicted output of the model, and $\omega_i^{(norm)}$ is the normalized weighting coefficient of the $i$-th sample. A regularization term is introduced in the loss function to prevent excessive attention to abnormal samples and increase the training weight on normal samples. The regularization term can use L2 norm regularization or other forms of regularization. The expression for the regularization term is:

$$\mathcal{L}_{reg} = \lambda \sum_{i=1}^{n} \left| \omega_i \right|^2$$

(11)

where $\lambda$ is the regularization coefficient, which controls the strength of the regularization term, $\omega_i$ is a parameter of the model, and $|\cdot|^2$ represents the L2 norm. In the optimization process, the CPLE algorithm measures the importance of abnormal samples by defining an abnormal sample metric. Assuming the prediction error of the sample $\in_i = |z_i - y_i|$, the anomaly measure is:

$$D_i = exp(-\alpha \cdot \in_i)$$

(12)

where $\alpha$ is the adjustment parameter for anomaly measurement, and $\in_i$ is the prediction error of the $i$-th sample. In the process of sample weighting, the final prediction results of the model can be adjusted through weighted averaging to enhance attention to abnormal samples. The weighted adjustment formula for the final prediction result is:

$$\hat{y}_i^{(adjusted)} = \frac{\sum_{j=1}^{n} \omega_j^{(norm)} \cdot \hat{y}_j}{\sum_{j=1}^{n} \omega_j^{(norm)}} \tag{13}$$

where $\omega_j^{(norm)}$ is the normalized weighting coefficient of the $j$-th sample, and $\hat{y}_j$ is the predicted value of the $j$-th sample. To enhance the influence on similar samples, a neighborhood influence measure between samples is defined $N_i$, which adjusts weights by calculating the similarity between samples and other samples. The calculation formula for neighborhood influence measurement is:

$$N_i = \sum_{j=1}^{n} exp(-\beta \cdot |z_i - z_j|) \tag{14}$$

where $z_j$ are the predicted results of samples, and $\beta$ is a hyperparameter that controls the range of influence measurement. To accelerate model convergence, the CPLE algorithm adjusts the dynamic learning rate by combining sample weights. Assuming the current learning rate is $\eta$, the update formula for the dynamic learning rate is:

$$\eta^{(t+1)} = \eta^{(t)} \cdot \left(1 + \gamma \cdot \sum_{i=1}^{n} \omega_i^{(norm)} \cdot \in_i\right) \tag{15}$$

where $\eta^{(t)}$ is the current learning rate, $\gamma$ is the learning rate adjustment rate, $\omega_i^{(norm)}$ is the normalized weighting coefficient of the sample, and $\in_i$ is the prediction error of the sample. In order to optimize the weighted loss function, the gradient update during backpropagation is adjusted based on the weighting coefficients of the samples. Assuming the gradient of the loss function is $\nabla \mathcal{L}$, the weighted gradient update formula is:

$$\nabla \mathcal{L}_{weighted} = \sum_{i=1}^{n} \omega_i^{(norm)} \cdot \nabla \mathcal{L}(y_i, \hat{y}_i) \tag{16}$$

where $\nabla \mathcal{L}(y_i, \hat{y}_i)$ is the standard gradient, and $\omega_i^{(norm)}$ is the weighting coefficient of the $i$-th sample. Through these measures, the CPLE algorithm can make the model more sensitive to detecting abnormal events and adapt more flexibly to the characteristics and prediction difficulty of different samples during the training process. To better understand the design of the Integrated Comparison Pessimistic Likelihood Estimation (CPLE) algorithm, we present it in pseudocode as shown in Algorithm 3.

## Algorithm 3. Heterogeneous graph neural network model

```
Algorithm 3 Integrated Comparison Pessimistic Likelihood Estimation (CPLE) Algorithm
Input: Training samples with predicted results z, true labels y, number of samples n, regularization
coefficient λ, adjustment parameters α, β, learning rate η
Output: Optimized model
// Update sample weights
1: for i = 1 to n do
2:     ωᵢ ← OriginalWeight(i)
3:     ωᵢ^(norm) ← ωᵢ/ sum(ωⱼ for j = 1 to n)
4: end for
// Adjust loss function
```

```
5: Weighted_loss←0
6: for i=1 to n do
7:     Loss_i ← StandardLossFunction(y_i, PredictedOutput_i)
8:     Weighted_loss←Weighted_loss + ω_i^(norm) * Loss_i
9: end for
10: Regularization_term ← λ * sum(ω_i^2 for i = 1 to n)
11: Total_loss←Weighted_loss+Regularization_term
// Calculate anomaly measure
12: for i = 1 to n do
13:     ∈_i ← |z_i − y_i|
14:     D_i ← exp(-α * ∈_i)
15: end for
// Adjust predictions
16: for i=1 to n do
17:     Adjusted_prediction_i←sum(ω_j^(norm) * PredictedOutput_j for j=1 to n)/ sum(ω_j^(norm) for j=1 to n)
18: end for
// Calculate neighborhood influence
19: for i=1 to n do
20:     N_i ← sum(exp(-β * |z_i − z_j|) for j = 1 to n)
21: end for
// Update learning rate
22: for t=1 to number_of_training_epochs do
23:     η^(t+1) ← η^t * (1+γ * sum(ω_i^(norm) * ∈_i for i=1 to n))
24: end for
// Adjust gradient update
25: for i = 1 to n do
26:     Gradient_i←StandardGradient(y_i, PredictedOutput_i)
27:     Weighted_gradient_i ← ω_i^(norm) * Gradient_i
28: end for
29: UpdateModelParameters(Weighted_gradient)
return Optimized_model
```

## Experiment and result analysis

### Dataset selection and experimental parameter setting and process

This study employed multiple real-world highway traffic datasets, encompassing diverse types of traffic anomaly events like car accidents, traffic congestion, and weather-induced changes. These datasets, sourced from different regions and environmental conditions, offer rich features and various patterns of traffic anomalies, making them ideal for analyzing and predicting traffic flow variations and abnormal events on highways.

The primary highway traffic datasets used in this research include:

(1) METR – LA dataset [traffic flow data of Los Angeles, https://github.com/liyaguang/DCRNN]: Originating from the highway network in the Los Angeles area, this dataset contains traffic flow information from multiple sensors. It covers indicators such as vehicle speed, traffic density, and flow rate, spanning multiple time periods. It is well-suited for predicting traffic flow and identifying abnormal events. Each sensor location samples data every five minutes, featuring traffic flow, vehicle speed, and traffic density, among other parameters. It includes traffic anomalies like traffic congestion, car accidents, and traffic repairs.

(2) PEMS – BAY dataset [Bay Area traffic flow data, https://github.com/liyaguang/DCRNN]: Derived from the traffic monitoring system in the San Francisco Bay Area, this dataset comprises a large amount of sensor data, covering traffic flow and vehicle speed information. It is suitable for traffic flow prediction, traffic pattern analysis, and anomaly detection. The data includes traffic flow, vehicle speed, and other indicators at multiple intersections, with a time interval of five minutes. It encompasses traffic anomalies such as fluctuations in traffic flow caused by weather changes, sudden accidents, and traffic control.

Given the diverse origins and characteristics of these datasets, several preprocessing steps were implemented to ensure the quality and consistency of the model input data:

(1) Data Cleaning and Noise Removal: The datasets were initially cleaned by eliminating corrupted or missing data points, which are common in real-world traffic sensor data. Anomalous sensor readings due to malfunctions were addressed by either interpolating or excluding outlier data points.

(2) Normalization and Standardization: As the datasets come from different regions (Los Angeles and the Bay Area) and may have varying scales (e.g., different units for vehicle speed or density), all numerical values were normalized to a common scale. Min – max normalization was applied to each feature (e.g., vehicle speed, traffic density, etc.) to transform the data into a uniform range (typically [0, 1]). This step ensures that the model learns without being biased towards any particular feature, enhancing convergence and stability during training.

(3) Time Alignment: Since the datasets have a time granularity of five – minute intervals, all data were synchronized to ensure temporal alignment across different regions. Time windows were adjusted as necessary to ensure consistent representation of traffic flow and anomaly events.

(4) Handling Missing Data: Missing data, a common challenge in traffic datasets, was managed using interpolation techniques. Linear interpolation was primarily used for short gaps, while more sophisticated methods (such as forward/backward filling) were employed for longer periods of missing data. This ensured the dataset remained continuous and reliable for model training.

(5) Feature Engineering and Integration: To enhance model performance, additional features were engineered. This involved calculating rolling averages and traffic density changes over different time windows to capture short – term fluctuations. Data from multiple sources were then integrated into a unified format, where each data point contained traffic information (flow, speed, density) for a specific region, time stamp, and sensor location. This integration merged the METR – LA and PEMS – BAY datasets into a common feature space, making them directly comparable.

(6) Anomaly Labeling: Traffic anomalies such as congestion, accidents, and weather – related disruptions were labeled based on known event logs or detected patterns in the data (e.g., sudden spikes in traffic flow). This enabled supervised training, allowing the model to recognize and predict similar events.

Through these preprocessing steps, the model's input data was standardized and cleaned, meeting the necessary quality and consistency requirements for accurate prediction and anomaly detection across different highway regions. The integrated dataset, after preprocessing, contained uniform and relevant traffic features from multiple regions, facilitating more robust analysis and reliable forecasting of traffic patterns and anomalies.

The training and validation sets were designed to ensure the model could effectively learn and predict in different traffic modes. The training set included traffic flow data from the METR – LA and PEMS – BAY datasets, which were normalized and time – aligned to cover features such as vehicle speed, flow, and density for multiple road segments and different time periods. Normal and abnormal data were balanced using oversampling and undersampling methods. The validation set was randomly selected from the raw data to ensure no duplication with the training set. It was mainly used to evaluate the model's generalization ability and cover various traffic modes. The ratio of the training set to the validation set was typically 80% and 20%, ensuring the model could fully learn during training and its performance could be effectively evaluated during validation. To optimize the model parameters and avoid overfitting, we used a validation subset separate from both the training and testing data. The parameters were tuned based on the validation performance. Specifically, we analyzed a range of values for each parameter, such as learning rates from 0.0001 to 0.01, batch sizes from 32 to 128, and dropout rates from 0.3 to 0.7, to find the optimal settings that maximize the model's accuracy on the validation set.

In the experiment, the dataset was first subjected to data cleaning and preprocessing to ensure data quality and consistency. The main preprocessing steps included filling missing traffic data through interpolation and the nearest neighbor method, detecting and processing outlier data points using statistical methods, standardizing all features to make the data suitable for deep learning models, and dividing the data into time series according to experimental requirements to generate training and testing sets. The ratio of the training set to the test set was 80:20 to ensure data representativeness and the model's generalization ability. The specific model parameter settings are presented in Table 2.

## Comparison model and evaluation indicator selection

In the experimental simulation, apart from simulating the MHGNN – CPLE model algorithm proposed in this paper, two advanced models were selected as comparison models: the Graph Convolutional Long Short Term Memory Network (AGC – LSTM) with a self – attention mechanism and the Attention Deep Aggregation Model (AttentionDeepST) with a self – attention mechanism. AGC – LSTM integrates graph convolution and LSTM models and incorporates a self – attention mechanism, enabling it to effectively track relevant features during the fusion process and improve the accuracy of the algorithm model. The AttentionDeepST model adopts a multi – level depth model for fusion and a cross – mode self – attention mechanism correction, making the model more accurate and stable and capable of meeting the fusion processing requirements in complex scenarios. This paper compares these two algorithm models to verify the correctness and progressiveness of the algorithm design.

To comprehensively evaluate the performance of this framework, four commonly used evaluation metrics were employed: accuracy [33], recall [34], F1 score [35], and AUC (area under the curve) [36]. By comparing these indicators, we can comprehensively understand the performance differences between our framework and traditional methods in detecting abnormal traffic events on highways from different dimensions, providing a basis for optimizing and improving subsequent models.

## Experimental results and performance analysis

1) The algorithm model in this article is used to detect and predict abnormal events in both static and dynamic scenarios

**Table 2. Experimental model parameter settings.**

| Parameter | Set value | Explain |
|---|---|---|
| Figure Neural Network Layers | 3 levels | Set the number of layers in the graph neural network to extract spatial dependencies in the graph structure. |
| Learning rate | 0.001 | Initial learning rate, using Adam optimizer. |
| Batch Size | 64 | The number of samples input during each training session. |
| CPLE algorithm sample weighting strategy | Linear weighting | The sample weighting strategy uses a linear weighting scheme to enhance the influence of different types of samples. |
| Time window length | 10 time units (5 minutes/unit) | The sliding window length used for time series data. |
| Training epochs | 100 rounds | The total number of rounds of network training until the model converges. |
| Model regularization parameters | 0.0001 | L2 regularization parameter used to prevent overfitting. |
| Activation function | ReLU | Activation function used for hidden layers to enhance non-linear fitting ability. |
| Hidden layer size | 128 neurons | The number of neurons in the hidden layer of a graph neural network. |
| Dropout rate | 0.5 | Dropout ratio used to prevent overfitting. |
| Nonlinear effect modeling method | GCN combined with LSTM (dual-mode) | Combining Graph Convolutional Networks (GCN) and Long Short Term Memory Networks (LSTM) to model nonlinear interaction effects. |
| objective function | MSE (Mean Squared Error) | Error measurement used for regression tasks to optimize the accuracy of model predictions. |

In the process of analysis, this article first takes the analysis of abnormal traffic congestion caused by a traffic accident in a tunnel as an example. The traffic flow information inside the tunnel is collected by image acquisition sensors, as shown in Fig 3. Firstly, through simple vehicle detection and distance recognition, the risk of abnormal traffic congestion in the current area is detected.

From the detection results in the above figure, it can be seen that the current traffic flow information in the tunnel has been abnormal through a single video image information, but it cannot determine the specific abnormal event. Therefore, this article will fuse the real-time average speed, traffic flow, and road accident rate of the corresponding time period to form feature data that covers real-time video image features, traffic environment features, and other information, as shown in Fig 4.

Based on the processing of the above feature data, the fused feature data was used to detect and predict abnormal events in highway traffic. The data was analyzed for normal, extreme weather, congestion, and accident scenarios, as shown in Fig 5.

By predicting and analyzing traffic abnormal events in these three scenarios, and integrating information on weather, traffic flow, and average speed measurement of different roads, the occurrence rates of different types of accidents in various scenarios were analyzed. The results are shown in Table 3.

From the above experimental results, it can be seen that the model can detect and predict various types of traffic abnormal events on highways based on the fused feature data. The experimental results show that the model can effectively identify and predict multiple types of traffic abnormal events. In the comparison between actual event types and predicted results, the model has a higher prediction accuracy. For traffic congestion, the predicted event types of the model are highly consistent with the actual events, and the predicted probability of anomalies is not significantly different from the actual probability. For example, when the vehicle speed is 40.5 km/h and the traffic flow is 150.2 vehicles, the actual and predicted probabilities of traffic congestion events are 0.85 and 0.83, respectively. For traffic accidents and weather related events, the model can also accurately predict, and the difference between the prediction and the actual situation is within an acceptable range, demonstrating good recognition ability. Under normal traffic conditions, the model has a low probability of predicting abnormal events, which is consistent with the actual situation. For example, when the vehicle speed is 60.4 km/h and the flow rate is 130.3 vehicles, the predicted and actual abnormal probabilities are 0.27 and 0.25, respectively. In summary, the model that integrates feature data has shown high accuracy and stability in the detection and prediction of traffic anomalies.

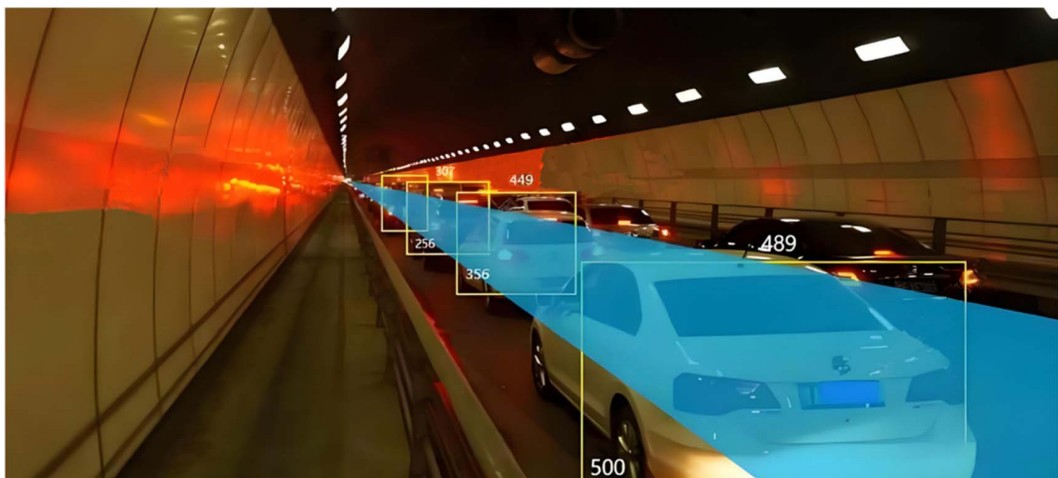

**Fig 3. Monitoring of image and video feature information under traffic congestion in tunnel.**

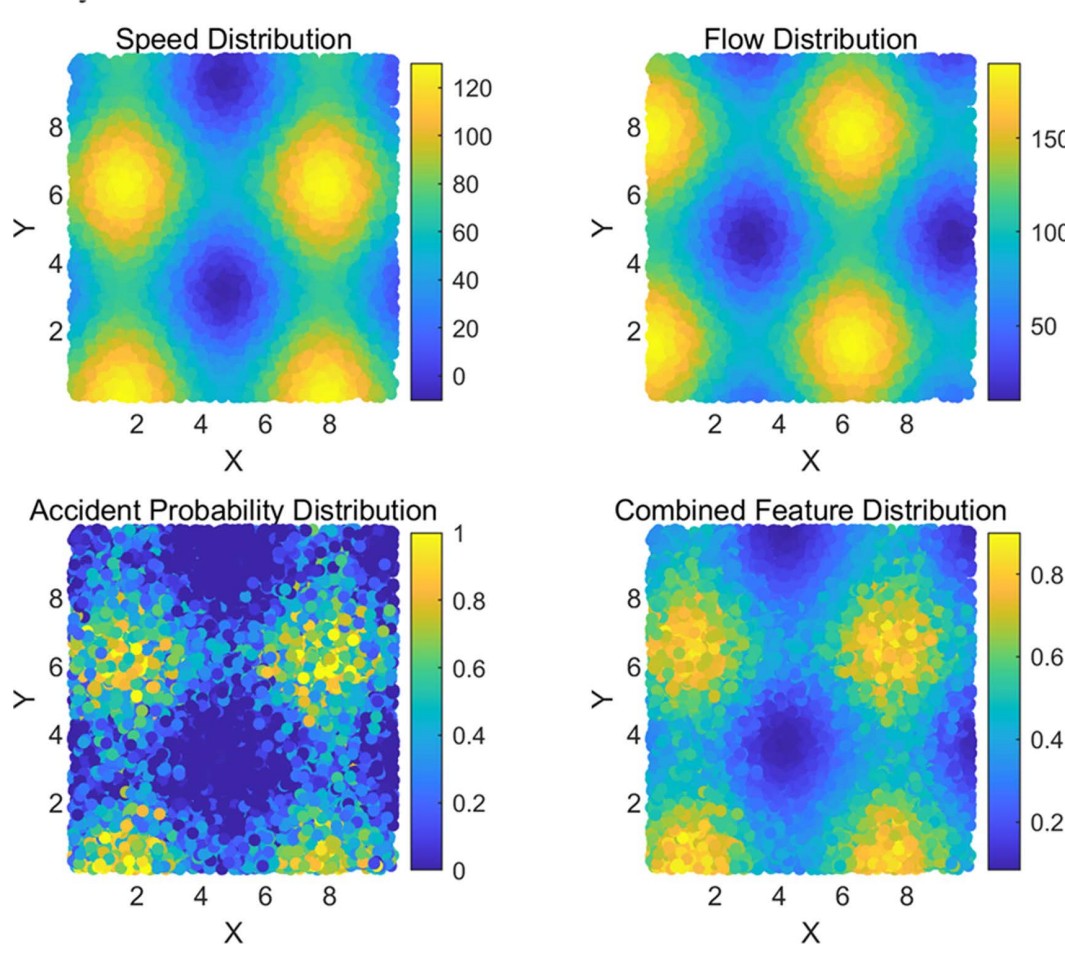

**Fig 4. Fused feature data.**

From the above detection results, it can be seen that the algorithm proposed in this paper can well meet the detection and analysis of abnormal events on highways for static processing. In order to further verify the dynamic nature of the algorithm proposed in this paper, the input video data stream and time series information of sensors are used to achieve dynamic detection of traffic abnormal information on highway sections. The monitoring feature perception results of different scene dimensions of the system input and the fused dynamic information obtained first are shown in Fig 6.

The results of pulse detection in four traffic event scenarios can be seen from Fig 3. In the "normal traffic" scenario, the signal changes smoothly, and there is no significant fluctuation in abnormal pulse detection, with a peak fluctuation amplitude of about 0.3. In the "accident" scenario, a clear pulse appeared in the signal at 40 seconds with an amplitude of 0.5, and the detection algorithm successfully marked the abnormal pulse. In the scenario of "traffic congestion", the signal exhibits periodic fluctuations with an amplitude of about 0.3, and the detected abnormal pulses are relatively frequent, but there are many false alarms. In the "weather impact" scenario, the signal presents a low-frequency cosine waveform with amplitude fluctuations around 0.2, indicating accurate anomaly detection but slightly larger errors. Through simulations with different noise levels (noise range 0.05 to 0.3), it can be seen that as the noise level increases, the accuracy of anomaly detection gradually decreases, especially in high noise environments. Although the detected pulse error increases, the increase is small, indicating that the algorithm model in this paper has strong robustness to noise

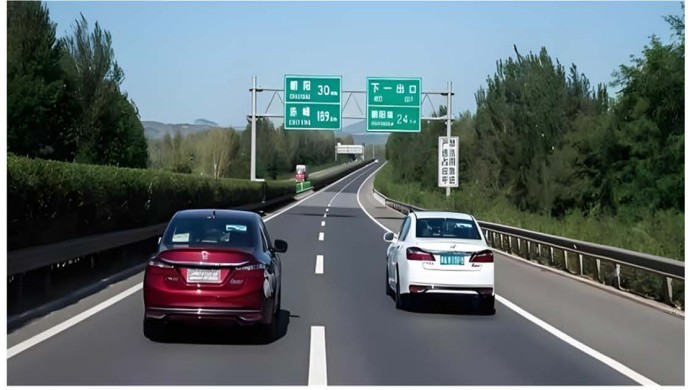

Normal

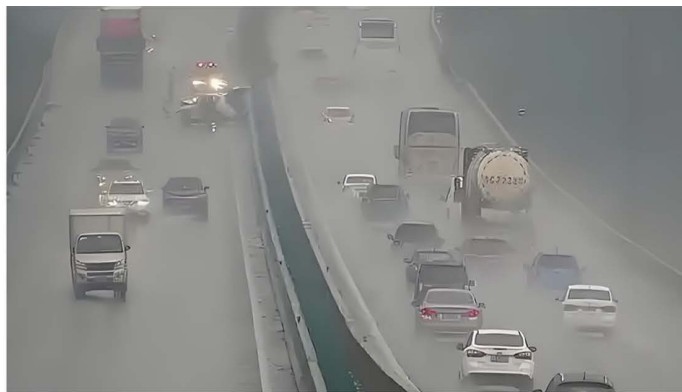

Abnormal weather conditions

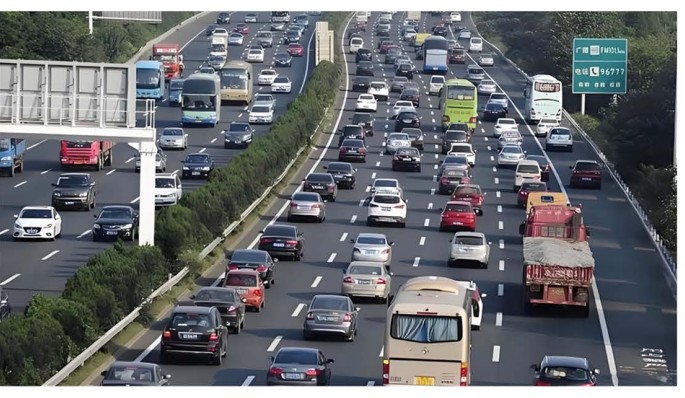

Abnormal congestion

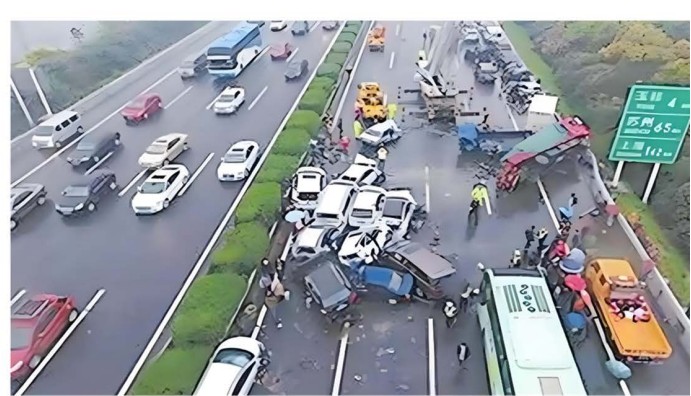

Abnormal accident

**Fig 5. Typical prediction analysis scenario.**

**Table 3. Prediction results of traffic abnormal events under different fusion information inputs.**

| Speed (km/) | Flow (vehicles) | Weather (Index) | Time of Day (Index) | Actual Event Type | Actual Anomaly Probability | Predicted Event Type | Predicted Anomaly Probability |
|---|---|---|---|---|---|---|---|
| 40.5 | 150.2 | 0.80 | 0.9 | Traffic Congestion | 0.85 | Traffic Congestion | 0.83 |
| 20.3 | 90.7 | 0.90 | 0.2 | Accident | 0.92 | Accident | 0.91 |
| 60.4 | 130.3 | 0.75 | 0.6 | Normal Traffic | 0.25 | Normal Traffic | 0.27 |
| 35.8 | 175.0 | 0.70 | 0.8 | Traffic Congestion | 0.78 | Traffic Congestion | 0.79 |
| 50.2 | 100.1 | 0.85 | 0.4 | Normal Traffic | 0.18 | Normal Traffic | 0.20 |
| 5.0 | 50.0 | 1.00 | 0.3 | Weather-related | 0.90 | Weather-related | 0.88 |
| 15.2 | 95.3 | 0.95 | 0.1 | Accident | 0.91 | Accident | 0.89 |
| 55.0 | 120.0 | 0.70 | 0.7 | Normal Traffic | 0.30 | Normal Traffic | 0.32 |
| 80.3 | 160.2 | 0.60 | 0.5 | Normal Traffic | 0.22 | Normal Traffic | 0.23 |
| 10.5 | 180.1 | 0.85 | 0.9 | Traffic Congestion | 0.80 | Traffic Congestion | 0.82 |

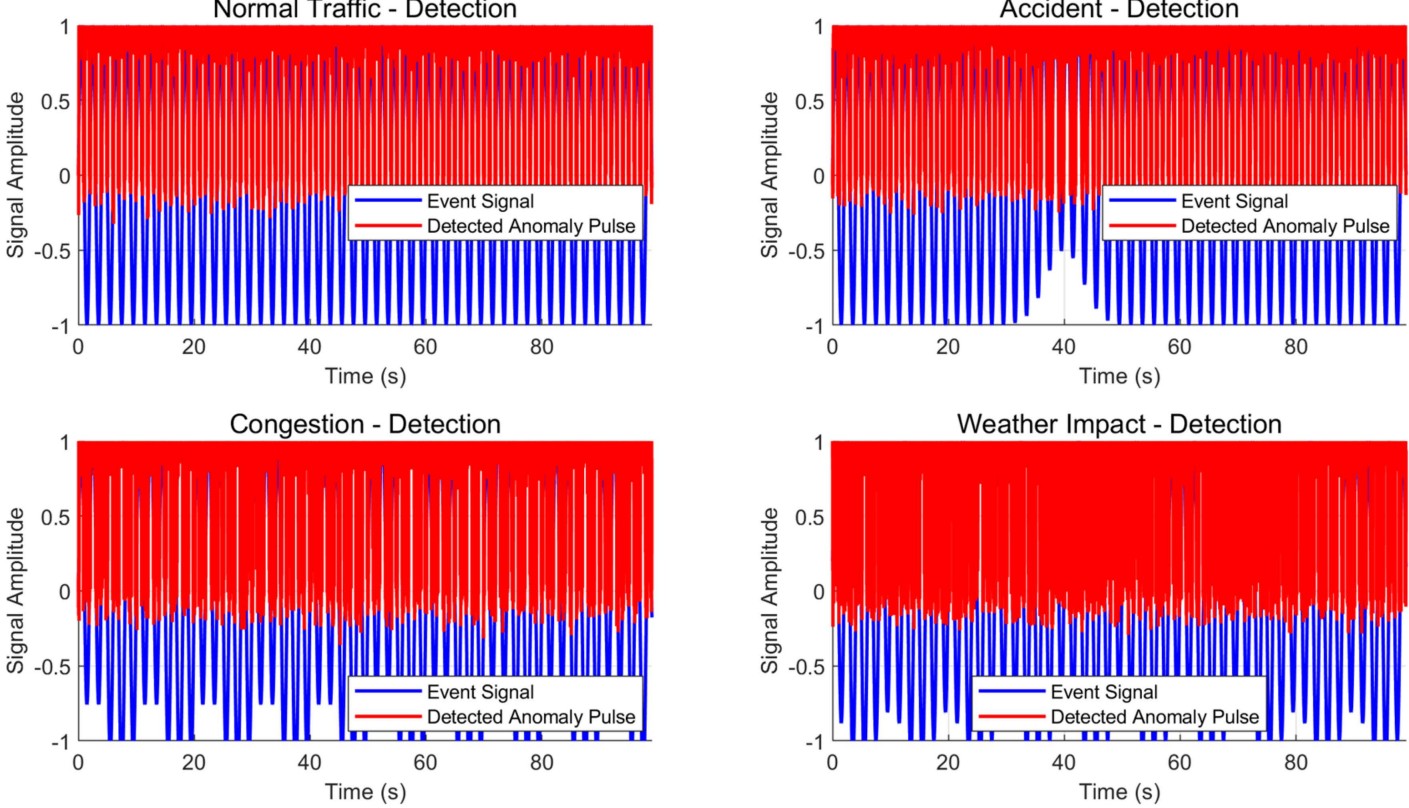

**Fig 6. Pulse signal diagram of dynamic abnormal event detection results in four different scenarios.**

interference in real-time detection algorithms. Table 4 shows the dynamic detection results of various indicators obtained from 1000 experiments.

At the same time, in order to observe the stability of various performance indicators in different scenarios in the time dimension, this paper monitored and analyzed the data in the experiment, detecting the changes in performance parameters of various scenarios in the time dimension, as shown in Figs 7–10. The changes under normal traffic scenarios are shown in Fig 7.

The changes in the time dimension of various indicators in the accident scenario are shown in Fig 8.

The changes in various indicators in the time dimension under congestion scenarios are shown in Fig 9.

The changing trends of various indicators in the time dimension under the influence of adverse weather conditions are shown in Fig 10.

**Table 4. Overall performance indicators of the algorithm model in this article for dynamic detection of traffic anomalies on highways.**

| Event scenario | Accuracy | Precision | Recall rate | F1 score | AUC (Area Under Curve) |
|---|---|---|---|---|---|
| Normal traffic | 0.965 | 0.980 | 0.950 | 0.965 | 0.985 |
| accident | 0.960 | 0.980 | 0.940 | 0.960 | 0.975 |
| traffic jam | 0.950 | 0.965 | 0.930 | 0.960 | 0.970 |
| Weather impact | 0.955 | 0.975 | 0.930 | 0.952 | 0.980 |

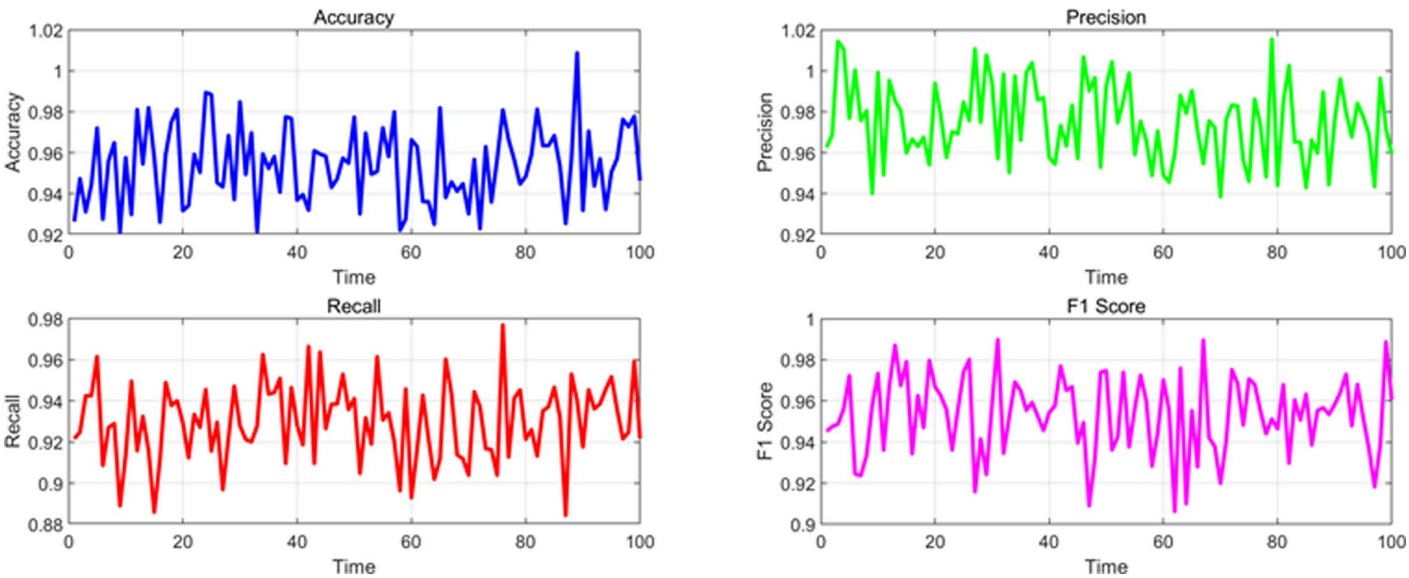

**Fig 7. Changes in various performance indicators in the time dimension under normal scenarios.**

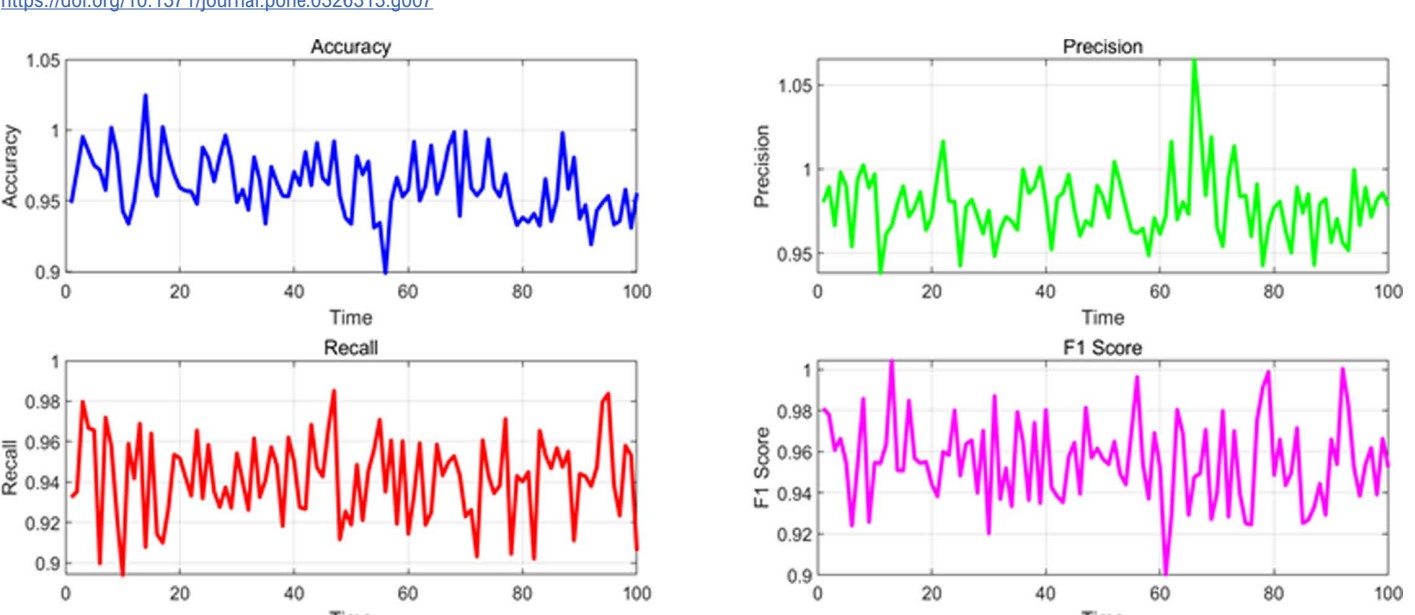

**Fig 8. Changes in various performance indicators in the time dimension under accident scenarios.**

From the above experimental results, it can be seen that the overall performance indicators of the algorithm model in dynamic detection of traffic anomalies on highways in this paper. For different event scenarios, the model exhibits high performance in accuracy, precision, recall, F1 score, and AUC (Area Under Curve) under normal traffic, accidents, congestion, and weather conditions. In normal traffic scenarios, the accuracy of the model is 0.965, the precision is 0.980, the recall is 0.950, the F1 value is 0.965, and the AUC is 0.985, all of which demonstrate excellent detection capabilities. The indicators for accident and congestion scenarios are slightly lower, but still maintain a high level,

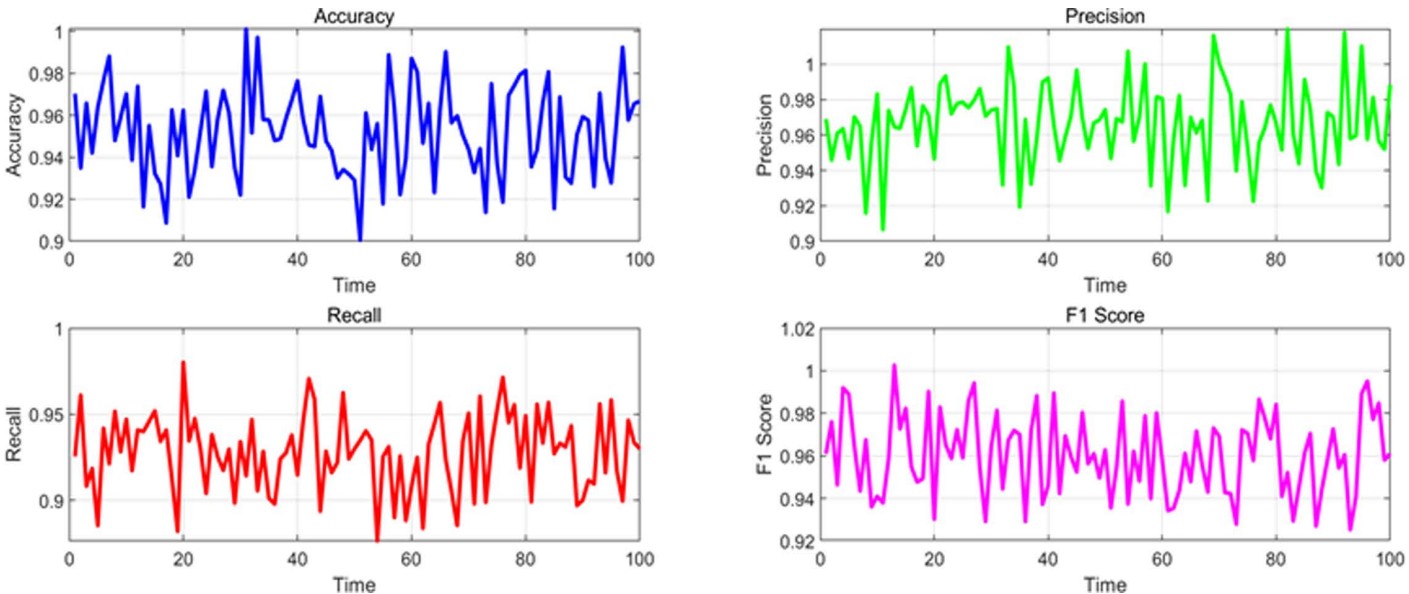

**Fig 9. Changes in various performance indicators in the time dimension under congested scenarios.**

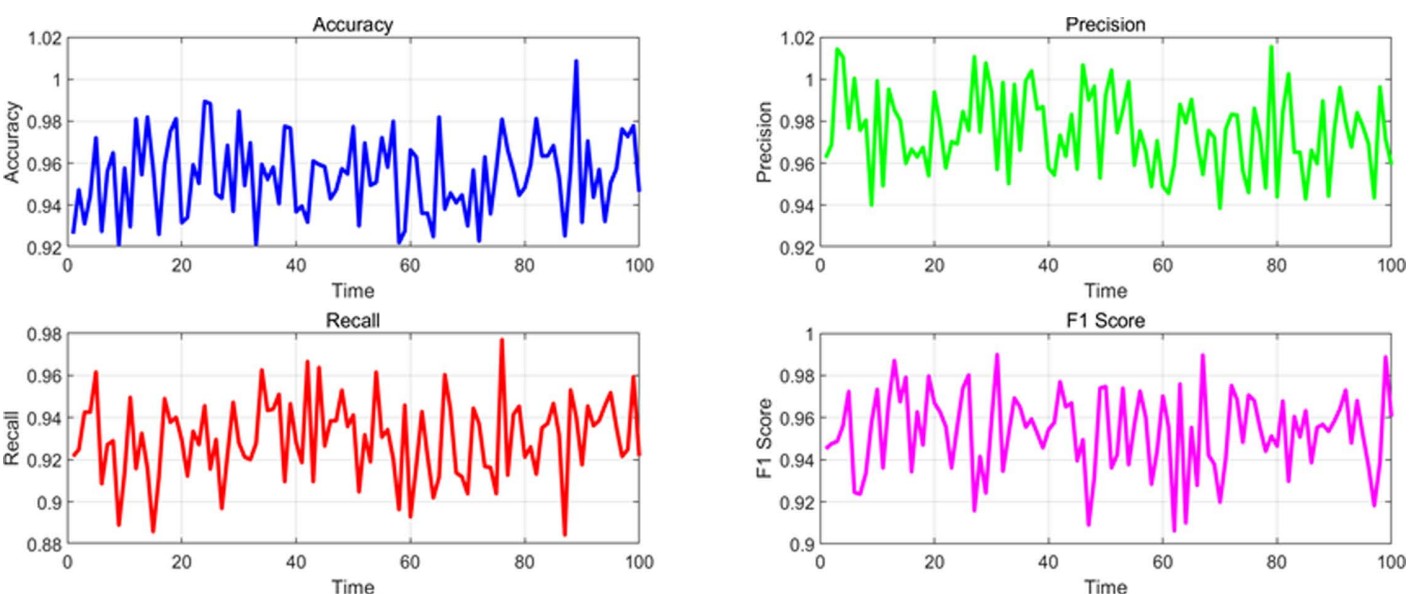

**Fig 10. Changes in various performance indicators in the time dimension under the influence of severe weather conditions.**

especially with accuracy and F1 values close to 0.96. Under the influence of weather conditions, the performance of the model is relatively stable, with various indicators approaching 0.95 or above, indicating that the algorithm model has good robustness and reliability in different traffic event scenarios, and can effectively perform dynamic traffic anomaly detection.

2) Comparative experimental analysis with existing algorithm models

In order to verify the progressiveness of the algorithm in this paper compared with the comparison algorithm model, in the process of experimental simulation, this paper compared and analyzed the designed MHGNN-CPLE model algorithm, the graph convolution long short memory network (AGC-LSTM) of the self attention mechanism, and the deep agglomeration model (AttentionDeepST) of the self attention mechanism.

Firstly, considering the traffic flow scenario, the static detection task aims to evaluate the performance of different models in identifying and classifying traffic flow data. Static detection tasks generally require models to accurately determine traffic conditions (such as normal, congested, etc.) during specific time periods. This experiment evaluates the accuracy and reliability of three models, MHGNN-CPLE, AGC-LSTM, and AttentionDeepST, by comparing their performance in static detection tasks. The results are shown in Table 5.

From the experimental results, it can be seen that MHGNN-CPLE performs well in static detection tasks, with an accuracy of 0.980, far higher than the other two models. The accuracy of AGC-LSTM is 0.950, with a slightly lower recall rate, indicating that there may be some missed detections during the detection process. In contrast, although AttentionDeepST has a higher accuracy (0.960), its precision and recall are lower than MHGNN-CPLE, resulting in a lower F1 value (0.935). Overall, MHGNN-CPLE not only leads in detection accuracy, but also has an AUC value of up to 0.985, demonstrating its robustness and accuracy in processing traffic flow data.

And in static scenarios, in traffic flow prediction, the model needs to effectively model historical data and make accurate predictions based on data patterns. This experiment compared the performance of MHGNN-CPLE, AGC-LSTM, and AttentionDeepST in static prediction tasks, and the results are shown in Table 6.

From the experimental results, MHGNN-CPLE has the best predictive performance, with an accuracy of 0.975 and high precision and recall, demonstrating its powerful ability in traffic flow prediction. The AUC value of 0.980 further proves its stability and accuracy in actual prediction. In contrast, the performance of AGC-LSTM is relatively balanced, with an accuracy of 0.950, but slightly lacking in accuracy and recall. The F1 value is 0.932, indicating that it may be slightly conservative in handling traffic flow prediction. AttentionDeepST performs poorly in static prediction tasks, with a difference in accuracy and recall resulting in an F1 value of 0.920. The overall prediction performance is not as good as the first two, as shown in Fig 11.

Secondly, considering that in dynamic scenarios, the main task is to identify and detect accidents. In accident detection scenarios, the core of dynamic detection tasks is to timely identify the moment when accidents occur and classify them. This type of task requires the model to be able to quickly respond in real-time changing traffic environments. By comparing the performance of three models in accident detection tasks, we evaluated their accuracy, recall rate, and F1 score, and the results are shown in Table 7.

**Table 5. Comparison of static detection performance (Traffic flow scenario).**

| Model algorithm | Accuracy | Precision | Recall rate | F1 score | AUC (Area Under Curve) |
|---|---|---|---|---|---|
| MHGNN-CPLE | 0.980 | 0.975 | 0.960 | 0.967 | 0.985 |
| AGC-LSTM | 0.965 | 0.950 | 0.940 | 0.945 | 0.960 |
| AttentionDeepST | 0.960 | 0.940 | 0.930 | 0.935 | 0.950 |

**Table 6. Comparison of static prediction performance (Traffic flow prediction scenario).**

| Model algorithm | Accuracy | Precision | Recall rate | F1 score | AUC (Area Under Curve) |
|---|---|---|---|---|---|
| MHGNN-CPLE | 0.975 | 0.960 | 0.950 | 0.955 | 0.980 |
| AGC-LSTM | 0.950 | 0.945 | 0.920 | 0.932 | 0.965 |
| AttentionDeepST | 0.940 | 0.930 | 0.910 | 0.920 | 0.960 |

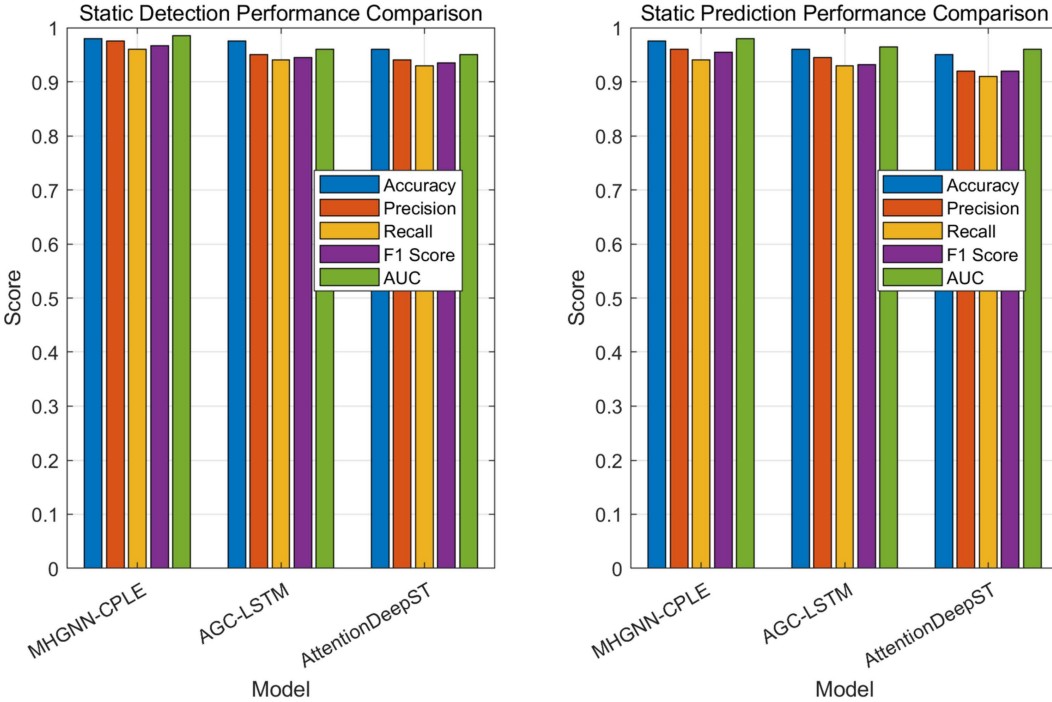

**Fig 11. Comparison of detection and prediction performance of different algorithm models in static scenes.**

**Table 7. Comparison of dynamic detection performance (Accident detection scenario).**

| Model algorithm | Accuracy | Precision | Recall rate | F1 score | AUC (Area Under Curve) |
|---|---|---|---|---|---|
| **MHGNN-CPLE** | 0.950 | 0.960 | 0.930 | 0.945 | 0.970 |
| **AGC-LSTM** | 0.940 | 0.950 | 0.910 | 0.930 | 0.960 |
| **AttentionDeepST** | 0.920 | 0.920 | 0.890 | 0.905 | 0.950 |

The experiment shows that MHGNN-CPLE continues to demonstrate its advantages in dynamic detection, with an accuracy of 0.950, precision of 0.960, recall of 0.930, F1 value of 0.945, AUC of 0.970, and outstanding performance in all indicators. Relatively speaking, although the performance of AGC-LSTM is good, its recall rate is slightly insufficient (0.910), which may result in some accidents not being detected in a timely manner. The performance of AttentionDeepST in this task is relatively mediocre, with an accuracy of 0.920 and a recall of 0.890. The overall effect is relatively conservative, and there may be many missed reports. Therefore, the all-round advantages of MHGNN-CPLE make it perform more outstandingly in accident detection tasks.

Dynamic prediction tasks play an important role in traffic accident prediction, requiring models to analyze data in real-time and predict potential accidents that may occur in the future. The challenge of this task is that the model needs to have strong time series modeling capabilities to cope with fluctuations in traffic flow. Through this experiment, we compared the performance of three models in dynamic accident prediction tasks, and the results are shown in Table 8.

In accident prediction scenarios, the MHGNN-CPLE model performs the best with the highest performance indicators, especially in terms of accuracy (0.940), precision (0.950), recall (0.920), and AUC (0.965), demonstrating excellent predictive ability. In addition, its robustness was rated as "high", significance as "very significant", and robustness as "very high", indicating that the model can maintain excellent stability and reliability in different data environments.In contrast, although

**Table 8. Comparison of dynamic prediction performance (Accident prediction scenarios).**

| Model algorithm | Accuracy | Precision | Recall Rate | F1 Score | AUC | Robustness | Significance | Stability |
|---|---|---|---|---|---|---|---|---|
| **MHGNN-CPLE** | 0.940 | 0.950 | 0.920 | 0.935 | 0.965 | High | Very Significant | Very High |
| **AGC-LSTM** | 0.925 | 0.930 | 0.900 | 0.915 | 0.950 | Medium | Significant | High |
| **AttentionDeepST** | 0.910 | 0.900 | 0.880 | 0.890 | 0.930 | Medium | Moderate | Medium |

the AGC-LSTM model performs slightly worse with an accuracy of 0.925, precision of 0.930, recall of 0.900, F1 score of 0.915, and AUC of 0.950, it still demonstrates strong performance. Its robustness is "moderate", significance is "significant", and robustness is "high", indicating that the model performs well in most cases, but may be slightly inadequate in certain specific contexts.The AttentionDeepST model ranks last among all indicators, with an accuracy of 0.910, precision of 0.900, recall of 0.880, F1 score of 0.890, and AUC of 0.930, indicating relatively low predictive performance. Its robustness is "moderate", significance is "moderate", and robustness is "moderate", indicating that the model may not perform as well as the first two when dealing with complex and variable data.Overall, MHGNN-CPLE is the most competitive model among the three, suitable for application scenarios that require high accuracy and stability. AGC-LSTM is suitable for most applications, but may not perform as well as MHGNN-CPLE under extreme or complex conditions. AttentionDeepST is suitable for simpler application scenarios, where accuracy requirements are more relaxed The comprehensive comparison is shown in Fig 12.

The experimental results show that the proposed framework based on multimodal deep fusion and HGNN exhibits higher accuracy and robustness in the detection and prediction of traffic abnormal events than traditional methods. Especially in complex traffic scenarios, the fusion of multimodal data significantly improves the model's ability to capture detailed information. HGNN performs outstandingly in handling spatiotemporal dependencies, while the introduction of CPLE algorithm further enhances the model's sensitivity to abnormal events. Compared with existing methods, this framework can maintain high performance even in complex and noisy situations.

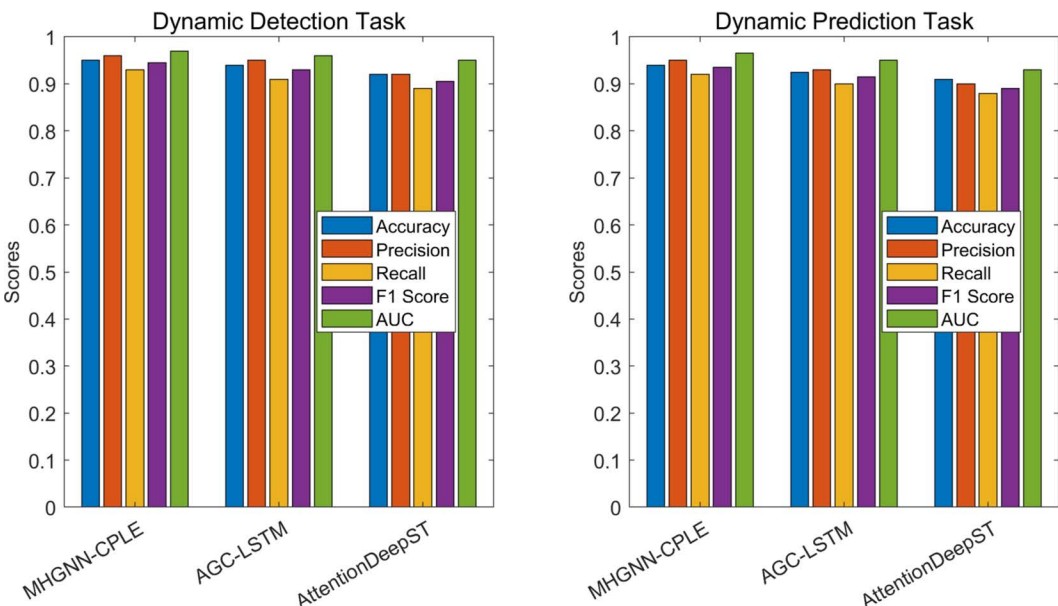

**Fig 12. Performance Comparison of Dynamic Detection and Prediction in Different Scenarios.**

3) Cross-Dataset Generalization Experimen

To comprehensively evaluate the generalization capability of the proposed framework, cross-dataset experiments were conducted. The model was trained on one dataset and tested on another unseen dataset to assess its adaptability to different traffic conditions and environments. In this experiment, the METR-LA dataset (Los Angeles traffic data) was used for training, while the PEMS-BAY dataset (San Francisco Bay Area traffic data) was employed for testing. The evaluation metrics used were accuracy, precision, recall, F1 score, and AUC. The results of the cross-dataset generalization experiment are presented in Table 9.

The experimental results indicate that the MHGNN-CPLE model demonstrates superior generalization performance across different datasets. It achieves an accuracy of 0.945 and an F1 score of 0.940 on the unseen PEMS-BAY dataset, significantly outperforming AGC-LSTM and AttentionDeepST. This highlights the proposed framework's ability to effectively adapt to new traffic environments and maintain high detection accuracy without requiring extensive retraining on the target dataset. The AGC-LSTM model also shows reasonable generalization ability but with slightly lower performance metrics compared to MHGNN-CPLE. Its accuracy and F1 score are 0.920 and 0.915, respectively. Meanwhile, the AttentionD-eepST model achieves the lowest performance among the three, with an accuracy of 0.905 and an F1 score of 0.895. This suggests that the AttentionDeepST model may not capture the underlying patterns and variations in unseen data-sets as effectively as the other models. Overall, the cross-dataset generalization experiment underscores the robustness and adaptability of the proposed MHGNN-CPLE framework. Its ability to generalize well across different traffic datasets makes it a promising solution for real-world highway traffic anomaly detection and prediction scenarios, where diverse and dynamic traffic conditions are prevalent. This generalization capability is crucial for deploying reliable and scalable traffic management systems that can operate effectively across various regions and traffic environments.

4) Model Robustness Testing Experiment

To further verify the robustness of the proposed model under different noise levels, we designed a model robustness testing experiment. In traffic anomaly event detection, data is often subject to various types of noise interference, such as sensor failures and environmental changes. Therefore, the stability of the model in noisy environments is crucial. In the experiment, we added Gaussian noise of different intensities to the original dataset to simulate data pollution sce-narios that may be encountered in practical applications. The noise intensities were set at 5%, 10%, 15%, and 20%. By comparing the detection performance of the models under different noise conditions, we assessed their robustness. The experiment used the PEMS-BAY dataset, and the evaluation metrics included accuracy, precision, recall, F1 score, and AUC.In the experiment, we added Gaussian noise of different intensities to the original dataset to simulate data pollution scenarios that may be encountered in practical applications. The noise intensities were set at 5%, 10%, 15%, and 20%. By comparing the detection performance of the models under different noise conditions, we assessed their robustness. The experiment used the PEMS-BAY dataset, and the evaluation metrics included accuracy, precision, recall, F1 score, and AUC. he results of the model robustness testing are presented in Table 10.

The experimental results show that as the noise intensity increases, the performance of all models decreases to some extent. However, the MHGNN-CPLE model demonstrates stronger noise resistance. At a noise intensity of 5%, the MHGNN-CPLE model achieves an accuracy of 0.960, an F1 score of 0.960, and an AUC of 0.980. Even at a high noise

Table 9. Cross-dataset generalization performance.

| Model Algorithm | Accuracy | Precision | Recall | F1 Score | AUC |
|---|---|---|---|---|---|
| MHGNN-CPLE | 0.945 | 0.950 | 0.930 | 0.940 | 0.970 |
| AGC-LSTM | 0.920 | 0.930 | 0.900 | 0.915 | 0.955 |
| AttentionDeepST | 0.905 | 0.910 | 0.880 | 0.895 | 0.945 |

**Table 10. Model robustness testing results.**

| Noise Intensity | Model Algorithm | Accuracy | Precision | Recall | F1 Score | AUC |
|---|---|---|---|---|---|---|
| 5% | MHGNN-CPLE | 0.960 | 0.970 | 0.950 | 0.960 | 0.980 |
| | AGC-LSTM | 0.945 | 0.955 | 0.935 | 0.945 | 0.975 |
| | AttentionDeepST | 0.935 | 0.945 | 0.925 | 0.935 | 0.965 |
| 10% | MHGNN-CPLE | 0.955 | 0.965 | 0.945 | 0.955 | 0.975 |
| | AGC-LSTM | 0.940 | 0.950 | 0.925 | 0.937 | 0.970 |
| | AttentionDeepST | 0.930 | 0.940 | 0.915 | 0.927 | 0.960 |
| 15% | MHGNN-CPLE | 0.950 | 0.960 | 0.940 | 0.950 | 0.970 |
| | AGC-LSTM | 0.935 | 0.945 | 0.920 | 0.932 | 0.965 |
| | AttentionDeepST | 0.925 | 0.935 | 0.910 | 0.922 | 0.955 |
| 20% | MHGNN-CPLE | 0.945 | 0.955 | 0.935 | 0.945 | 0.965 |
| | AGC-LSTM | 0.930 | 0.940 | 0.915 | 0.927 | 0.960 |
| | AttentionDeepST | 0.920 | 0.930 | 0.905 | 0.917 | 0.950 |

intensity of 20%, the MHGNN-CPLE model still maintains an accuracy of 0.945, an F1 score of 0.945, and an AUC of 0.965. In contrast, the performance of the AGC-LSTM and AttentionDeepST models declines more 显著. For instance, at a noise intensity of 20%, the AGC-LSTM model has an accuracy of 0.930 and an F1 score of 0.927, while the AttentionDeepST model has an accuracy of 0.920 and an F1 score of 0.917. This indicates that the MHGNN-CPLE model can perform stably across different noise levels, showing better robustness to data noise. It is therefore more suitable for the complex data environments encountered in practical traffic anomaly event detection.

5) Ablation Study

To comprehensively evaluate the contributions of different data sources and model components to the performance of our proposed framework, an ablation study was conducted. The study systematically investigates the impact of various data source combinations and the influence of each model component by progressively removing them and observing the changes in model performance.

The ablation experiments were carried out on the same highway traffic datasets used in the previous experiments. Different combinations of data sources, including single data sources, combinations of two or three data sources, and the full set of data sources (video images, traffic flow, vehicle speed, and tunnel weather conditions), were utilized to train and test the model. Additionally, the impact of different model components, such as the multimodal data fusion mechanism, heterogeneous graph neural networks (HGNNs), and the Ensemble Contrastive Pessimistic Likelihood Estimation (CPLE) algorithm, was assessed by excluding each component one at a time.

The results of the ablation experiments are presented in Table 11. They reveal that the model's performance gradually improves as more data sources are integrated. Specifically, the model utilizing all four data sources achieves the highest accuracy of 0.980 and an F1 score of 0.967. In contrast, models relying on a single data source demonstrate significantly lower performance. For instance, the model using only video images attains an accuracy of 0.850 and an F1 score of 0.835, while the model using solely traffic flow data reaches an accuracy of 0.870 and an F1 score of 0.855. The combination of video images and traffic flow data proves to be particularly effective, with the model achieving an accuracy of 0.920 and an F1 score of 0.905. Further enhancements are observed when vehicle speed and tunnel weather conditions are added to the data mix.

Regarding the model components, the experiments demonstrate that both the multimodal data fusion mechanism and HGNNs are vital for optimal model performance. Removing either component leads to a notable decline in accuracy and F1 score. For example, eliminating the multimodal data fusion mechanism reduces the accuracy to 0.910 and the

**Table 11. Results of ablation experiments.**

| Experiment Configuration | Accuracy | Precision | Recall | F1Score | AUC |
|---|---|---|---|---|---|
| Video Images Only | 0.850 | 0.845 | 0.835 | 0.835 | 0.860 |
| Traffic Flow Only | 0.870 | 0.865 | 0.855 | 0.855 | 0.875 |
| Vehicle Speed Only | 0.820 | 0.815 | 0.805 | 0.805 | 0.830 |
| Tunnel Weather Only | 0.780 | 0.775 | 0.760 | 0.760 | 0.790 |
| Video + Traffic Flow | 0.920 | 0.915 | 0.905 | 0.905 | 0.925 |
| Video + Vehicle Speed | 0.890 | 0.885 | 0.875 | 0.875 | 0.895 |
| Video + Tunnel Weather | 0.860 | 0.855 | 0.845 | 0.845 | 0.870 |
| Traffic Flow + Vehicle Speed | 0.880 | 0.875 | 0.865 | 0.865 | 0.885 |
| Traffic Flow + Tunnel Weather | 0.840 | 0.835 | 0.825 | 0.825 | 0.850 |
| Vehicle Speed + Tunnel Weather | 0.810 | 0.805 | 0.795 | 0.795 | 0.820 |
| Video + Traffic Flow + Vehicle Speed | 0.940 | 0.935 | 0.925 | 0.925 | 0.945 |
| Video + Traffic Flow + Tunnel Weather | 0.930 | 0.925 | 0.915 | 0.915 | 0.935 |
| All Data Sources | 0.980 | 0.975 | 0.967 | 0.967 | 0.985 |
| Without Multimodal Fusion | 0.910 | 0.905 | 0.895 | 0.895 | 0.915 |
| Without HGNNs | 0.900 | 0.895 | 0.880 | 0.880 | 0.905 |
| Without CPLE Algorithm | 0.930 | 0.925 | 0.910 | 0.910 | 0.935 |

F1 score to 0.895. Similarly, the removal of HGNNs results in an accuracy of 0.900 and an F1 score of 0.880. The CPLE algorithm also plays a crucial role in bolstering the model's robustness and accuracy. Excluding the CPLE algorithm causes the accuracy to drop to 0.930 and the F1 score to 0.910.

In summary, the ablation study underscores the significance of integrating diverse data sources and leveraging advanced model components, including multimodal data fusion, HGNNs, and the CPLE algorithm, to achieve high accuracy and robustness in detecting and predicting abnormal traffic events on highways.

## Discussion and future work

The integration of multimodal deep fusion and HGNNs represents a significant advancement in the detection and prediction of highway traffic anomalies. By utilizing diverse data sources and combining HGNNs with the CPLE algorithm, this approach outperforms traditional methods. The multimodal data integration provides a comprehensive view of traffic scenarios, while HGNNs effectively model spatiotemporal dependencies. The CPLE algorithm further improves the model's sensitivity to abnormal events. However, practical applications face challenges such as data quality and consistency issues due to varying noise levels, missing values, and inconsistent formats across diverse data sources. Additionally, the model's high computational complexity arising from multiple data modalities and complex architectures needs to be addressed to meet real-time processing requirements. Furthermore, while exploring new data sources like sensor networks, social media, and vehicle-to-vehicle communications offers potential, their integration raises concerns about privacy, security, and compatibility. Future research should focus on developing advanced data cleaning and normalization techniques, enhancing model computational efficiency through methods such as compression and hardware acceleration, and validating the framework in various real-world scenarios. Addressing these challenges will pave the way for more intelligent traffic management systems, thereby improving road safety and traffic flow.

## Conclusion

The framework for detecting and predicting abnormal traffic events on highways, which integrates multimodal deep fusion and heterogeneous graph neural networks (HGNNs), effectively consolidates multiple data sources. By harnessing the

spatiotemporal modeling capabilities of graph neural networks alongside the optimization techniques of the CPLE algorithm, it substantially enhances the accuracy and robustness of detecting traffic anomalies. This approach has shown great potential in boosting detection performance. Nonetheless, practical implementation still faces several challenges, such as data quality, inconsistencies, and the demand for real-time processing. Given the highly dynamic and time-sensitive nature of traffic conditions, it is crucial to overcome these limitations to achieve optimal performance. Future research should focus on several key directions: refining data fusion techniques to better handle diverse and incomplete data, boosting the computational efficiency of models to satisfy real-time processing needs, and delving into additional data sources, like sensor networks, social media data, and vehicle – to – vehicle communications, to gain a more all – encompassing view of traffic events. These advancements are expected to give rise to more precise, scalable, and intelligent traffic management systems, which in turn will greatly enhance road safety and traffic flow.

## Acknowledgments

We are grateful to the reviewers for their constructive comments and suggestions, which helped improve the quality of this manuscript. Special thanks go to the Intelligent Transportation Engineering Institute and the School of Mechanical Engineering at Hefei University of Technology for providing the necessary research facilities and resources. We appreciate the contributions of our colleagues and research assistants for their valuable insights and support throughout the project. The authors would also like to acknowledge the data providers and contributors of the publicly available traffic datasets used in this study. Their efforts in collecting and sharing such valuable data have significantly enhanced the comprehensiveness and reliability of our research findings. Lastly, we extend our appreciation to our families for their unwavering support and encouragement, which have been instrumental in the successful completion of this work.

## Author contributions

**Funding acquisition:** Shaowei Sun.

**Methodology:** Shaowei Sun.

**Project administration:** Mingzhou Liu.

**Validation:** Shaowei Sun.

**Visualization:** Mingzhou Liu.

**Writing – original draft:** Shaowei Sun.

**Writing – review & editing:** Shaowei Sun.

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
