## [Decision Letter · Decision Letter 0]

Dear Dr. Sun,

Thank you for submitting your manuscript to PLOS ONE. After careful consideration, we feel that it has merit but does not fully meet PLOS ONE’s publication criteria as it currently stands. Therefore, we invite you to submit a revised version of the manuscript that addresses the points raised during the review process.

We look forward to receiving your revised manuscript.

Kind regards,

Jinlei Zhang 

Academic Editor

PLOS ONE

“The authors would like to express their sincere gratitude to the Science and Technology Project of Housing and Urban-Rural Construction of Anhui Province (Grant No. 2022-YF044) for their financial support, which enabled the research presented in this paper. We are also grateful to the reviewers for their constructive comments and suggestions, which helped improve the quality of this manuscript.

Special thanks go to the Intelligent Transportation Engineering Institute and the School of Mechanical Engineering at Hefei University of Technology for providing the necessary research facilities and resources. We appreciate the contributions of our colleagues and research assistants for their valuable insights and support throughout the project.

The authors would also like to acknowledge the data providers and contributors of the publicly available traffic datasets used in this study. Their efforts in collecting and sharing such valuable data have significantly enhanced the comprehensiveness and reliability of our research findings.

Lastly, we extend our appreciation to our families for their unwavering support and encouragement, which have been instrumental in the successful completion of this work.”

5. We note that your Data Availability Statement is currently as follows: [All relevant data are within the manuscript and its Supporting Information files.]

Reviewers' comments:

Reviewer's Responses to Questions

**Comments to the Author**

1. Is the manuscript technically sound, and do the data support the conclusions?

Reviewer #1: Yes

Reviewer #2: Yes

2. Has the statistical analysis been performed appropriately and rigorously?

Reviewer #1: Yes

Reviewer #2: Yes

3. Have the authors made all data underlying the findings in their manuscript fully available?

Reviewer #1: Yes

Reviewer #2: Yes

4. Is the manuscript presented in an intelligible fashion and written in standard English?

Reviewer #1: Yes

Reviewer #2: Yes

Reviewer #1: Suggest:

This paper addresses the problem that traditional traffic detection models usually rely on single data sources. The authors propose a new framework for detecting and predicting abnormal traffic events on highways. The method uses multimodal deep fusion and heterogeneous graph neural networks to effectively fuse multi-source heterogeneous traffic data. It also applies the CPLE algorithm to improve performance. As a result, the accuracy of real-time abnormal event detection is improved. The research is novel and meaningful, especially for handling heterogeneous traffic data. The experiments are also rich and detailed. However, the current version still does not meet the journal's standards. Some problems need to be fixed. The main issues are as follows:

1. The title is too long. Please make it shorter and focus on the key point of the research.

2. The abstract is also too long. It only needs to briefly mention the research background, main idea, and general experimental results. There is no need to list detailed results like accuracy or F1 scores of different models.

3. The logic in the section Introduction is not very clear. It is better to first introduce the research background, then review existing work and point out its limitations. After that, present your own model. This will make the structure more logical. At the end of the introduction, please summarize the main contributions so readers can understand the paper more easily.

4. In the related work section, it would be helpful to add a table comparing existing highway abnormal event detection models with the proposed model. This will better show the advantages of your method.

5. Figure 1 shows the model framework, which includes data collection and fusion, model development and evaluation, and model optimization. But the beginning of Section 3 does not match this structure. Please check this part and revise it carefully.

6. There are some errors in lines 175–176, 183–185, 193–195, and 198–199. Some parameter letters are missing. Please correct these sentences and carefully check the whole text to correct spelling mistakes.

7. The paper uses multimodal deep fusion and heterogeneous graph neural networks to detect and predict highway abnormal traffic events. It is suggested to add ablation experiments in the experimental section. You can compare different data source combinations to see how they affect the model's performance. You should also study how each part of the model affects the results.

8. Related work is quite incomplete. Many related SOTA traffic prediction studies are not reviewed in this paper.

Zhang J, Mao S, Yang L, et al. Physics-informed deep learning for traffic state estimation based on the traffic flow model and computational graph method[J]. Information Fusion, 2024, 101: 101971.

Zhang S, Zhang J, Yang L, et al. Physics Guided Deep Learning-based Model for Short-term Origin-Destination Demand Prediction in Urban Rail Transit Systems Under Pandemic[J]. Engineering, 2024.

Zhang J, Mao S, Zhang S, et al. EF-former for short-term passenger Flow Prediction during large-scale events in Urban Rail Transit systems[J]. Information Fusion, 2025, 117: 102916.

Zhang J, Zhang S, Zhao H, et al. Multi-frequency spatial-temporal graph neural network for short-term metro OD demand prediction during public health emergencies[J]. Transportation, 2025: 1-23.

Qiu H, Zhang J, Yang L, et al. Spatial–temporal multi-task learning for short-term passenger inflow and outflow prediction on holidays in urban rail transit systems[J]. Transportation, 2025: 1-30.

Reviewer #2: • The manuscript should have a section to describe state-of-the-art techniques. This section should also outline a tabular sketch so that it is easy to identify what’s missing in the literature and how this paper addresses that. This section can be derived from contents described in the introduction section.

• Authors should pattern the motivation behind using this method to explain in the introduction.

• The abstract is not coherent. It would be good if authors can write a sentence describing numerical results and improvement over other methods.

• The integration of diverse data types (video, speed, flow, weather) strengthens detection accuracy and robustness. However, the fusion strategy (early, late, or hybrid fusion) should be clearly explained, including how modalities are synchronized and weighted.

• There needs to be citation of recent papers on this topic and revise the literature section with slight Incorporation of recent ideas. Add more recent works in related section (Resilience to deception attacks in consensus tracking control of incommensurate fractional-order power systems via adaptive RBF neural network, Energy‐Efficient Resource Allocation for Urban Traffic Flow Prediction in Edge‐Cloud Computing, Fuzzy adaptive control for consensus tracking in multiagent systems with incommensurate fractional-order dynamics: Application to power systems, An Attention-Driven Spatio-Temporal Deep Hybrid Neural Networks for Traffic Flow Prediction in Transportation Systems, A resource-aware multi-graph neural network for urban traffic flow prediction in multi-access edge computing systems, A data aggregation based approach to exploit dynamic spatio-temporal correlations for citywide crowd flows prediction in fog computing)

• The CPLE algorithm appears to address uncertainty or imbalance in anomaly prediction, which is a common challenge. However, the intuition behind CPLE, its objective function, and how it contrasts with standard loss functions need elaboration.

Were any temporal GNNs (like ST-GCN or T-GCN) considered?

• Pattern the motivation behind using this method to explain in the introduction. Why the existing schemes failed? Does no study try to address this aspect before? If yes, this has to be mentioned.

• Parameters of network have been enhanced using training data "until the model obtains the maximum accuracy". If this accuracy is the training accuracy, maybe over-fitting has been performed. If this accuracy is the testing accuracy, the system is adjusted over the same subset that is evaluated. A validation subset could be used to optimize the system with different data than the testing data and without performing over-fitting. In addition, it would be interesting to know which range of each parameter has been analyzed."?

**Do you want your identity to be public for this peer review?** For information about this choice, including consent withdrawal, please see our Privacy Policy

Reviewer #1: No

Reviewer #2: **Yes: ** AMIN SHARAFIAN

---

## [Author Response · Author response to Decision Letter 1]

22 May 2025

Dear Editorial Department of PLOS ONE:

We would like to thank you for your review of our paper "A framework for detecting and predicting abnormal traffic events on highways based on multimodal deep fusion and heterogeneous graph neural networks: a method of integrating and comparing pessimistic likelihood estimation algorithms" and for your valuable review comments. We have carefully read and considered each review comment, and have made changes to the paper, and the changes have been highlighted. Below are our point-by-point responses to the review comments:

Response: Thank you for your reminder. We have carefully reviewed and updated our manuscript to ensure it meets the PLOS ONE style requirements. We have also double-checked the file naming conventions and made the necessary adjustments. If there are any further specific requirements or adjustments needed, please let us know.

2.Please note that PLOS ONE has specific guidelines on code sharing for submissions in which author-generated code underpins the findings in the manuscript. In these cases, we expect all author-generated code to be made available without restrictions upon publication of the work. Please review our guidelines at https://journals.plos.org/plosone/s/materials-and-software-sharing#loc-sharing-code and ensure that your code is shared in a way that follows best practice and facilitates reproducibility and reuse.

Response: Thank you for your reminder. We have reviewed the PLOS ONE guidelines on code sharing and have made the author-generated code publicly available via a GitHub repository. The code can be accessed using the following link: https://github.com/ShaoweiSun/A-framework-for-detecting-and-predicting-highway-traffic-anomalies-.git. We have also included the necessary documentation and licensing information to facilitate reproducibility and reuse.

3.We note that the grant information you provided in the ‘Funding Information’ and ‘Financial Disclosure’ sections do not match. When you resubmit, please ensure that you provide the correct grant numbers for the awards you received for your study in the ‘Funding Information’ section.

Response: Thank you for bringing this to our attention. We have reviewed the funding information and ensured that the correct grant numbers are provided in the ‘Funding Information’ section. The grant number for the study is 2022-YF044 from the Science and Technology Project of Housing and Urban-Rural Construction of Anhui Province.

“The authors would like to express their sincere gratitude to the Science and Technology Project of Housing and Urban-Rural Construction of Anhui Province (Grant No. 2022-YF044) for their financial support, which enabled the research presented in this paper. We are also grateful to the reviewers for their constructive comments and suggestions, which helped improve the quality of this manuscript.

Special thanks go to the Intelligent Transportation Engineering Institute and the School of Mechanical Engineering at Hefei University of Technology for providing the necessary research facilities and resources. We appreciate the contributions of our colleagues and research assistants for their valuable insights and support throughout the project.

The authors would also like to acknowledge the data providers and contributors of the publicly available traffic datasets used in this study. Their efforts in collecting and sharing such valuable data have significantly enhanced the comprehensiveness and reliability of our research findings.

Lastly, we extend our appreciation to our families for their unwavering support and encouragement, which have been instrumental in the successful completion of this work.”

Response: Thank you for your guidance. We have removed the funding-related text from the Acknowledgments section. Please update the Funding Statement to read: "This work was supported by the Science and Technology Project of Housing and Urban-Rural Construction of Anhui Province, Grant No. 2022-YF044."

5. We note that your Data Availability Statement is currently as follows: [All relevant data are within the manuscript and its Supporting Information files.]

Response: Thank you for your guidance. We confirm that our submission contains all the raw data required to replicate the study's results. The minimal data set, including the values behind the means, standard deviations, and graph - building values, is available within the manuscript and its Supporting Information files. We have also ensured compliance with PLOS ONE's definition of the minimal data.

Reviewer #1: Suggest:

This paper addresses the problem that traditional traffic detection models usually rely on single data sources. The authors propose a new framework for detecting and predicting abnormal traffic events on highways. The method uses multimodal deep fusion and heterogeneous graph neural networks to effectively fuse multi-source heterogeneous traffic data. It also applies the CPLE algorithm to improve performance. As a result, the accuracy of real-time abnormal event detection is improved. The research is novel and meaningful, especially for handling heterogeneous traffic data. The experiments are also rich and detailed. However, the current version still does not meet the journal's standards. Some problems need to be fixed. The main issues are as follows:

1. The title is too long. Please make it shorter and focus on the key point of the research.

Response: Thank you for your valuable feedback regarding the title of our manuscript. We agree that the original title was quite lengthy and could benefit from simplification to better emphasize the core of our research. Based on your suggestion, we have revised the title to "A Framework for Detecting and Predicting Highway Traffic Anomalies via Multimodal Fusion and Heterogeneous Graph Neural Networks." The modified content is as follows:

A Framework for Detecting and Predicting Highway Traffic Anomalies via Multimodal Fusion and Heterogeneous Graph Neural Networks

2. The abstract is also too long. It only needs to briefly mention the research background, main idea, and general experimental results. There is no need to list detailed results like accuracy or F1 scores of different models.

Response: Thanks for your feedback. We've revised the abstract to focus on the research background, main idea, and general results, removing specific metrics. The modified content is as follows:

Abstract

This paper presents a novel framework for detecting and predicting abnormal traffic events on highways. Current traffic monitoring systems often rely on single data sources, which limits their detection accuracy and robustness in complex environments. To address these limitations, we propose a framework based on multimodal deep fusion and heterogeneous graph neural networks (HGNNs), incorporating an Ensemble Contrastive Pessimistic Likelihood Estimation (CPLE) algorithm to optimize performance. The framework integrates static and dynamic traffic data, such as video images, traffic flow, vehicle speed, and tunnel weather conditions. Experimental results demonstrate that the model performs well in various scenarios, showing significant improvement in accuracy and stability over existing models like AGC-LSTM and AttentionDeepST. For instance, the proposed MHGNN-CPLE model achieves superior accuracy and F1 score in static detection tasks while maintaining high accuracy under different noise levels in dynamic detection scenarios. This study provides a significant advancement in traffic event analysis by effectively combining multimodal data and leveraging HGNNs to capture complex spatiotemporal dependencies.

3. The logic in the section Introduction is not very clear. It is better to first introduce the research background, then review existing work and point out its limitations. After that, present your own model. This will make the structure more logical. At the end of the introduction, please summarize the main contributions so readers can understand the paper more easily.

Response: Thank you for your feedback on the introduction section. We have revised the introduction to first present the research background, followed by a review of existing work and its limitations, and then introduce our proposed model. The main contributions are summarized at the end of the introduction. This new structure should provide a clearer and more logical flow for readers.

4. In the related work section, it would be helpful to add a table comparing existing highway abnormal event detection models with the proposed model. This will better show the advantages of your method.

Response: Thank you for your suggestion. We have revised the "Related Work" section to include a comparative table that highlights the differences between existing models for highway abnormal event detection and our proposed model. This table aims to clearly demonstrate the unique features and advantages of our approach.

5. Figure 1 shows the model framework, which includes data collection and fusion, model development and evaluation, and model optimization. But the beginning of Response: Section 3 does not match this structure. Please check this part and revise it carefully. Thank you for pointing out the inconsistency between the beginning of Section 3 and Figure 1. We have carefully revised Section 3 to align with the model framework presented in Figure 1. The modified content is as follows:

In this research, a framework for detecting and predicting abnormal traffic events on highways based on multimodal deep fusion and heterogeneous graph neural networks (HGNNs) is constructed. The aim is to overcome the limitations of traditional methods and enhance the accuracy and robustness of detection and prediction. This framework mainly encompasses three core components: data collection and fusion, model development and evaluation, and model optimization. The overall architecture is shown in Figure 1.

6. There are some errors in lines 175–176, 183–185, 193–195, and 198–199. Some parameter letters are missing. Please correct these sentences and carefully check the whole text to correct spelling mistakes.

Response: Thank you for your valuable comments. We have carefully addressed the issues in lines 175–176, 183–185, 193–195, and 198–199, and corrected the missing parameter letters. The revised equations and explanations have been integrated into the manuscript without altering the original structure. We have also conducted a thorough check to ensure no other spelling mistakes are present.

7. The paper uses multimodal deep fusion and heterogeneous graph neural networks to detect and predict highway abnormal traffic events. It is suggested to add ablation experiments in the experimental section. You can compare different data source combinations to see how they affect the model's performance. You should also study how each part of the model affects the results.

Response: Thank you for your suggestion. We have added an ablation study section to the experimental part of our paper. This section evaluates the contributions of different data sources and model components by systematically removing them and assessing the impact on model performance. The results demonstrate the importance of multimodal data fusion and each component of our model in achieving high accuracy and robustness in detecting and predicting abnormal traffic events.

8. Related work is quite incomplete. Many related SOTA traffic prediction studies are not reviewed in this paper.

Zhang J, Mao S, Yang L, et al. Physics-informed deep learning for traffic state estimation based on the traffic flow model and computational graph method[J]. Information Fusion, 2024, 101: 101971.

Zhang S, Zhang J, Yang L, et al. Physics Guided Deep Learning-based Model for Short-term Origin-Destination Demand Prediction in Urban Rail Transit Systems Under Pandemic[J]. Engineering, 2024.

Zhang J, Mao S, Zhang S, et al. EF-former for short-term passenger Flow Prediction during large-scale events in Urban Rail Transit systems[J]. Information Fusion, 2025, 117: 102916.

Zhang J, Zhang S, Zhao H, et al. Multi-frequency spatial-temporal graph neural network for short-term metro OD demand prediction during public health emergencies[J]. Transportation, 2025: 1-23.

Qiu H, Zhang J, Yang L, et al. Spatial–temporal multi-task learning for short-term passenger inflow and outflow prediction on holidays in urban rail transit systems[J]. Transportation, 2025: 1-30.

Response: Thank you for pointing out the incompleteness of the related work. We have revised the related work section to incorporate the recent studies you mentioned. We briefly introduce the key contributions of these works and how they relate to our research. We have also updated the comparison table in the state-of-the-art techniques section to reflect these new additions. This enhancement provides a more comprehensive overview of the current research landscape and better positions our contribution within it.

Reviewer #2: 

1.The manuscript should have a section to describe state-of-the-art techniques. This section should also outline a tabular sketch so that it is easy to identify what’s missing in the literature and how this paper addresses that. This section can be derived from contents described in the introduction section.

Response: Thank you for your suggestion. We have added a new section titled "Comparat

---

## [Editor Report · Decision Letter 1]

A framework for detecting and predicting highway traffic anomalies via multimodal fusion and heterogeneous graph neural networks

PONE-D-25-22834R1

Dear Dr. Sun,

We’re pleased to inform you that your manuscript has been judged scientifically suitable for publication and will be formally accepted for publication once it meets all outstanding technical requirements.

Kind regards,

Academic Editor

PLOS ONE
---

## [Editor Report · Acceptance letter]

PONE-D-25-22834R1

PLOS ONE

Dear Dr. Sun,

I'm pleased to inform you that your manuscript has been deemed suitable for publication in PLOS ONE. Congratulations! Your manuscript is now being handed over to our production team.

Kind regards,

on behalf of

Dr. Jinlei Zhang

Academic Editor

PLOS ONE